# RGB-Only Supervised Camera Parameter Optimization in Dynamic Scenes

**Fang Li**
University of Illinois at Urbana-Champaign
Champaign, IL 61820
fangli3@illinois.edu

**Hao Zhang**
University of Illinois at Urbana-Champaign
Champaign, IL 61820
haoz19@illinois.edu

**Narendra Ahuja**
University of Illinois at Urbana-Champaign
Champaign, IL 61820
n-ahuja@illinois.edu

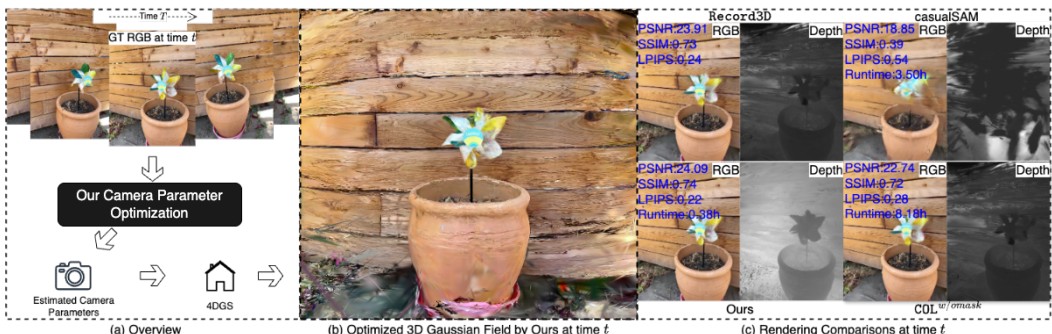

Figure 1: (a) Overview of our RGB-only supervised camera parameter optimization. (b) Front view of the **3D Gaussian field** reconstructed by our camera estimates at time $t$. (c) **2D renderings** (RGB and depth) at time $t$ with quantitative metrics. Our optimization is not only significantly more efficient and accurate, but also avoids overfitting the reconstruction to specific viewpoints. Record3D is *a mobile app* that factory-calibrates the intrinsic and uses LiDAR sensors to collect metric depth for camera pose estimates, thus does not have valid runtime.

## Abstract

Although COLMAP has long remained the predominant method for camera parameter optimization in static scenes, it is constrained by its lengthy runtime and reliance on ground truth (GT) motion masks for application to dynamic scenes. Many efforts attempted to improve it by incorporating more priors as supervision such as GT focal length, motion masks, 3D point clouds, camera poses, and metric depth, which, however, are typically unavailable in casually captured RGB videos. In this paper, we propose a novel method for more accurate and efficient camera parameter optimization in dynamic scenes solely supervised by a single RGB video, dubbed ***ROS-Cam***. Our method consists of three key components: (1) Patch-wise Tracking Filters, to establish robust and maximally sparse hinge-like relations across the RGB video. (2) Outlier-aware Joint Optimization, for efficient camera parameter optimization by adaptive down-weighting of moving outliers, without reliance on motion priors. (3) A Two-stage Optimization Strategy, to enhance stability and optimization speed by a trade-off between the Softplus limits and convex minima in losses. We visually and numerically evaluate our camera estimates. To

39th Conference on Neural Information Processing Systems (NeurIPS 2025).

further validate accuracy, we feed the camera estimates into a 4D reconstruction method and assess the resulting 3D scenes, and rendered 2D RGB and depth maps. We perform experiments on 4 real-world datasets (NeRF-DS, DAVIS, iPhone, and TUM-dynamics) and 1 synthetic dataset (MPI-Sintel), demonstrating that our method estimates camera parameters more efficiently and accurately with a single RGB video as the only supervision.

# 1 Introduction

Despite recent progress in visual odometry, efficiently and accurately optimizing camera parameters[1] (focal length + rotation&translation) from casually collected RGB dynamic-scene videos remains a big challenge. Although the most predominant COLMAP [32] method[2] is RGB-only supervised, it suffers from its lengthy runtime and requisite of GT motion masks to mask out the outlier moving stuff. In Table 1, most recent approaches [6, 45, 3, 42, 59, 46, 56, 44] attempted to improve through being supervised by additional GT priors such as focal length, metric depth, 3D point clouds, camera poses, and motion masks, which are typically unavailable in casually collected videos. We cannot help but ask a natural question: *Is it possible to accurately and efficiently estimate camera parameters in dynamic scenes in an RGB-only supervised manner - the most minimal form of supervision?*

Existing RGB-only supervised methods [42, 45, 20, 59, 3] make obvious improvements, but they mostly rely on multiple pre-trained dense prediction models [38, 13, 31] to compensate for the inaccuracies of individual pseudo-supervision sources, resulting in performance degradation if any of them fails. They also cannot adaptively exclude moving outliers without GT motion supervision. Besides, their high computational latency always leads to lengthy runtimes. Further discussion of related work is provided in Section 2.

Table 1: **Categorization of supervision of current methods.** Ours, casualSAM [58], and Robust-CVD [16] are RGB-only supervised, while our performance is the best as shown in section 4.

| Supervision | Static Scene | Dynamic Scene |
|---|---|---|
| **GT 3D Point Cloud & Camera Pose** | Dust3r [46], Fast3r [52], Mast3r [18], Spann3r [40], VGGT [41] | Monst3r [56], Cut3r [44], Stereo4D [11], Easi3r [4] |
| **GT Focal Length** *+ Metric Depth* *+ GT Motion Priors* | CF-3DGS [6], Nope-NeRF [1], LocalNeRF [22] | DROID-SLAM [39] GFlow [45], LEAP-VO [3] |
| **GT Motion Priors** | | RoDynRF [20], COL[w/ mask] [32], ParticleSfM [59] |
| **RGB-Only** | VGGSfM [42], FlowMap [35], InstantSplat [5], COL[w/o mask] [32] | Robust-CVD [16], casualSAM [58], **Ours (*ROS-Cam*)** |

Based on these insights, we propose ***ROS-Cam***, an RGB-only supervised, accurate, and efficient camera parameter optimization method, with a brief performance overview in Figure 1. Specifically, to minimize reliance on pre-trained dense prediction models while still establishing robust and maximally sparse hinge-like relations across the video as accurate pseudo-supervision (bottom right corner in Figure 2), we propose the novel patch-wise tracking filters built solely on a pre-trained point tracking (PT) model. This formulation effectively avoids inaccurate tracking trajectories extracted across frames and computational latency induced by the noisy dense prediction as pseudo-supervision.

However, the extracted pseudo-supervision includes a portion of trajectories belonging to moving outliers. To eliminate the influence of such outliers, we introduce a learnable uncertainty associated with each calibration point, where each is a learnable 3D position in the world coordinates, corresponding to one extracted tracking trajectory. We model such uncertainty parameters with the Cauchy distribution, which can deal with heavy tails better than, e.g., the Gaussian distribution, and propose the novel Average Cumulative Projection error and Cauchy loss for the outlier-aware joint optimization of the calibration points, focal length, rotation, translation, and uncertainty parameters. Unlike casualSAM and LEAP-VO, which assign uncertainty parameters to 2D pixels, our approach associates uncertainties with sparse 3D calibration points, resulting in significantly fewer learnable parameters and reduced runtime, as shown in Table 3.

Such joint optimization is prone to getting trapped in local minima. To address this, we analyze the asymptotic behavior of the Softplus function and the analytical minima of the inner convex term in

---

[1]Like all existing methods in table 1, we also assume a pinhole camera.

[2]We denote the COLMAP using motion masks as COL[w/ mask] and the one w/o motion masks as COL[w/o mask].

losses to propose a two-stage optimization strategy to accelerate and stabilize the optimization. We evaluate the performance of our method through extensive experiments on 5 popular public datasets - NeRF-DS [50], DAVIS [28], iPhone [7], MPI-Sintel [2], and TUM-dynamics [36], demonstrating our superior performance. Our contributions can be summarized as follows.

- We propose the first RGB-only supervised, accurate, and efficient camera parameter optimization method in dynamic scenes with three key components: (1) *patch-wise tracking filters*; (2) *outlier-aware joint optimization*; and (3) a *two-stage optimization strategy*.

- We present exhaustive quantitative and qualitative experiments and extensive ablation studies that demonstrate the superior performance of our proposed method and the contribution of each component.

## 2    Related Works

**Dynamic Scene Reconstruction/Novel View Synthesis (NVS).** Existing methods for reconstructing objects and scenes use a variety of 3D representations, including planar [8, 9], mesh [51, 55], point cloud [48, 57], neural field [23, 50, 37, 29, 21], and the recently introduced Gaussian explicit representations [49, 14, 10, 47, 53]. NeRF [23] enables high-fidelity NVS. Some methods [25, 26, 29, 50, 15, 24] also extend NeRF to dynamic scenes, while others [51, 55, 54, 43, 37] build on them to extract high-quality meshes. However, NeRF-based methods have the limitation of a long training time. Recently, 3DGS [14] effectively addressed this issue by using 3D Gaussian-based representations and presented Differential-Gaussian-Rasterization in CUDA. 3DGS optimizes 3D Gaussian ellipsoids as dynamic scene representations associated with attributes such as position, orientation, opacity, scale, and color. Several studies [47, 53] also have used 3DGS for dynamic scenes, achieving near real-time dynamic scene novel view synthesis. However, both NeRF-based and 3DGS-based methods heavily rely on $\texttt{COL}^{\text{w/ mask}}$ to estimate camera parameters.

**Camera Parameter Optimization.** Many efforts have been made to overcome the shortcomings of COLMAP, particularly for dynamic scenes. But each suffers from some constraints. In Table 1, we present a categorization of supervision of current SOTA methods. Supervised by additional GT focal length, CF-3DGS [6], Nope-NeRF [1], and LocalNeRF [22] leverage a pre-trained monocular depth estimation model [31] to estimate camera poses and the static scene jointly. The most representative SLAM-based method - DROID-SLAM [39], leverages both GT focal length and metric depth as supervision. GFlow and LEAP-VO [45, 3] extend it to dynamic scenes with both GT focal length and motion priors as supervision. Although VGGSfM, FlowMap, InstantSplat, $\texttt{COL}^{\text{w/o mask}}$ [42, 35, 5, 32] eliminate the GT focal length requirement by leveraging pre-trained PT models [38, 13], they cannot handle the moving objects in dynamic scenes. RoDynRF [20], $\texttt{COL}^{\text{w/ mask}}$ [32], and ParticleSfM [59] simply tackle such a problem by incorporating GT motion supervision like GFlow. Recently, DUSt3R-based methods [46, 52, 18, 40, 41] and their dynamic-scene counterparts [56, 44, 11, 4] explored feed-forward camera parameter prediction by training on large-scale static and dynamic scene datasets, respectively, **in a fully supervised manner** - that is, using GT 3D point clouds and camera poses as supervision, requiring several days' training on high-end GPUs. However, unlike LLMs that benefit from abundant language data, such metric 3D supervision is relatively scarce in the vision area, leading to frequent domain gaps when these models are applied to unseen data. In contrast, Robust-CVD [16], casualSAM [58] and our method conduct camera parameter optimization for dynamic scenes **in a more general RGB-only supervised way**. However, as shown in Section 4, their performance is significantly worse than ours.

## 3    Method

Under RGB-only supervision, RGB frames $F_i, i \in [0, N-1]$ ($N$ is frame count) are given. Our proposed patch-wise tracking filters (Section 3.1) extract $H$ robust and maximally sparse hinge-like tracking trajectories as pseudo-supervision, where each corresponds to one calibration point $\mathbf{P}_h^{cali} \in \mathbb{R}^3, h \in [0, H]$ in the world coordinates. Under such pseudo-supervision and our newly proposed ACP error and Cauchy loss, the calibration points $\mathbf{P}^{cali}$, focal length $f \in \mathbb{R}$, quaternion matrix $\mathbf{Q} \in \mathbb{R}^{N \times 4}$, translation $\mathbf{t} \in \mathbb{R}^{N \times 3}$, and motion-caused uncertainty parameters $\mathbf{\Gamma} \in \mathbb{R}_{>0}^H$ are jointly optimized (Section 3.2). $\mathbf{\Gamma}$ is the scale parameter of the Cauchy distribution which is used to model such uncertainty parameters, associated with each calibration point, to reduce the erroneous

influence of moving outliers. By analyzing the Softplus limits and convex minima in losses, we propose a simple but effective two-stage optimization strategy (Section 3.3) to enhance the stability and optimization speed.

## 3.1 Patch-wise Tracking Filters

Built on a pre-trained PT model, we observe that its attention mechanism assigns higher attention weights to pixels with more accurate tracking results which are always texture-rich pixels with large gradient norms. Inspired by it, as shown in Figure 2, we propose the patch-wise texture filter to identify the high-texture patches within $F_t$ and the patch-wise gradient filter to select the pixel with the highest gradient norm within each identified patch. While tracking such identified points, the visibility filter keeps removing trajectories that become invisible and the patch-wise distribution filter keeps the one with the largest gradient norm when multiple moving points enter the same patch. As shown in appendix E.2.1 (fig. 10), our method only retains the robust and accurate trajectories as pseudo-supervision.

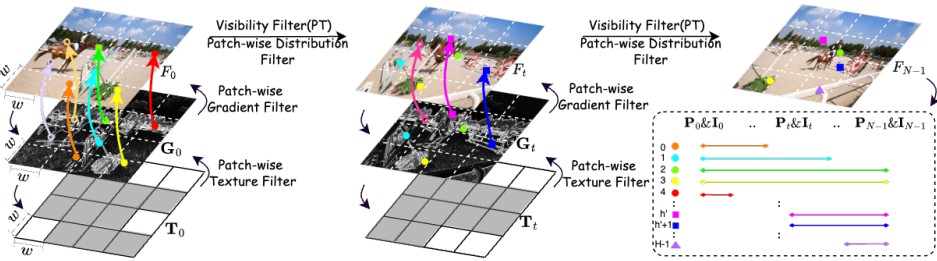

Figure 2: **Patch-wise tracking filters.** (1) Partitioning $F_0$ into patches of size $w \times w$, the patch-wise texture filter computes the texture map $\mathbf{T}_0$ and marks the high-texture patches in gray; (2) Within each high-texture patch, the patch-wise gradient filter selects one potential tracking point with the highest gradient norm. (3) The visibility filter removes the entire trajectory of a point if it becomes invisible at any time ($\bullet$, $\blacksquare$, $\blacktriangle$ → kept trajectories; $\circ$, $\square$, $\triangle$ → removed trajectories); (4) The patch-wise distribution filter only keeps the one with the largest gradient norm when multiple trajectories fall into the same patch. $\mathbf{P}$ and $\mathbf{I}$ are the location and index of trajectory, and $\leftrightarrow$ is the trajectory range.

**Patch-wise Texture Filter.** Highly distinguishable points, that can be tracked reliably, belong to highly nonuniform (textured) neighborhoods. To identify such neighborhoods, our patch-wise texture filter computes a texture map $\mathbf{T}_i \in \mathbb{1}^{\mathbb{H}/w \times \mathbb{W}/w}$, giving a measure of texture level for each $w \times w$ patch where $\mathbb{H}$ and $\mathbb{W}$ denote the height & width of $F_i$. We represent the texture level of a patch by

$$\mathbf{T}_i[m, n] = \mathbb{1}\{\Sigma_i[m, n] > \tau_{var} \cdot \sigma^*\} \tag{1}$$

, where $\Sigma_i \in \mathbb{R}^{\mathbb{H}/w \times \mathbb{W}/w}$ is the intensity variance, $\sigma^* = \max(\Sigma_i)$, $\tau_{var}$ is the percentage threshold of minimum variance for the patch to be selected, and $m, n \in [0, \mathbb{H}/w - 1], [0, \mathbb{W}/w - 1]$. The texture levels of a patch are represented by 1 for the selected patches and 0 for the others.

**Patch-wise Gradient Filter.** Within the identified patches, our patch-wise gradient filter computes the intensity gradient norm map $\mathbf{G}_i \in \mathbb{R}^{\mathbb{H} \times \mathbb{W}}$ of $F_i$, and selects the point with the largest gradient norm within each patch. This yields the pool of potentially distinguishable points, forming potential trajectories, namely,

$$\mathbf{P}_{m,n}^{potential} = \arg\max_p(\mathbf{G}_i[mw : mw + w, nw : nw + w]), \text{ p} \to \text{pixel locations} \tag{2}$$

**Visibility Filter.** We find that current PT models [13, 12, 30] still tend to suffer from reduced tracking accuracy when a point becomes occluded and later reappears, due to the disruption of temporal feature continuity. Thus, if any $\tilde{\mathbf{P}}$ in any $F_i$ becomes invisible, our visibility filter deletes it by the dot product $\tilde{\mathbf{P}} \cdot \mathbf{V}$, $\mathbf{V} \in \{0, 1\} \sim \tilde{\mathbf{P}}$, where $\mathbf{V} = 0$ if a point is invisible.

**Patch-wise Distribution Filter.** This filter enforces a more even point distribution within each frame, preventing them from clustering into a small region as the viewpoint changes. It also helps reduce susceptibility to loss of resolution which might result in triangulation errors. We keep the highest-gradient tracking point $\tilde{\mathbf{P}}^*$ in each patch $Pat_{m,n}$ of $F_i$, as follows:

$$\tilde{\mathbf{P}}^* = \arg\max_{\tilde{\mathbf{P}}} \mathbf{G}_i[\tilde{\mathbf{P}} \in Pat_{m,n}], \text{ if } \sum \mathbb{1}(\tilde{\mathbf{P}} \in Pat_{m,n}) > 1 \tag{3}$$

As shown in Figure 2, locations and indices of $\tilde{\mathbf{P}}^*$ are stored in $\mathbf{P}_i \in \mathbb{R}^{B \times 2}$ and $\mathbf{I}_i \in \mathbb{R}^B$, $i \in [0, H-1]$, acting as pseudo-supervision in the outlier-aware joint optimization. Each iteration starts at $F_t$, $t = \arg\min_t(-1 \in \mathbf{I}_t)$, and ends *until* each frame contains exactly $B$ tracked points.

## 3.2 Outlier-aware Joint Optimization

**Outlier-aware Joint Optimization Mechanism.** Under the obtained pseudo-supervision, $\mathbf{P}_{cali}$, $f$, $\mathbf{Q}$, $\mathbf{t}$ and $\mathbf{\Gamma}$ are jointly optimized. We first project $\mathbf{P}^{cali-homo} \in \mathbb{R}^{H \times 4}$ (the homogeneous coordinates of $\mathbf{P}^{cali}$ obtained by concatenating 1) onto each frame by

$$\mathbf{P}_i^{proj-homo} = \mathbf{P}^{cali-homo}[\mathbf{I}_i] \cdot \begin{bmatrix} \mathbf{R}_i & \mathbf{t}_i \\ \mathbf{0} & 1 \end{bmatrix}^T \cdot \mathbf{K}^T \tag{4}$$

$$\mathbf{P}_i^{proj} = \mathbf{P}_i^{proj-homo}[:, :2]/\mathbf{P}_i^{proj-homo}[:, 3] \tag{5}$$

, where $i \in [0, N-1]$ and $\mathbf{P}^{proj} \in \mathbb{R}^{N \times B \times 2}$. $\mathbf{P}^{proj-homo} \in \mathbb{R}^{N \times B \times 4}$ denotes the homogeneous 2D location of the projection $\mathbf{P}^{proj}$. The perspective projection matrix $\mathbf{K} \in \mathbb{R}^{4 \times 4}$ is derived from $f$, and the world-to-camera transformation matrix consists of rotation $\mathbf{R}_i$ and translation $\mathbf{t}_i$. We assume constant $f$ like SOTA [59, 58, 20]. Notably, we learn the quaternion matrix $\mathbf{Q}_i$ instead of optimizing the $\mathbf{R}_i$ and additional constraints. This optimization approach circumvents the difficult-to-enforce orthogonality and $\pm 1$ determinant constraints required for rotation matrices during optimization.

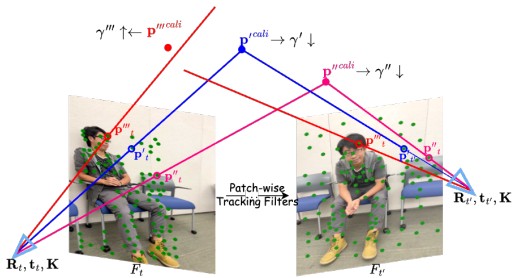

Figure 3: **Outlier-aware Joint Optimization.** • represents $\mathbf{P}_t$ and $\mathbf{P}_{t'}$ on each frame. The static samples $p'^{cali}$ and $p''^{cali}$ can establish concrete triangulation relations with their corresponding $\mathbf{P}_t$, $\mathbf{P}_{t'}$, and cameras, resulting in lower $\gamma'$ and $\gamma''$. In contrast, the dynamic sample $p'''^{cali}$ exhibits the opposite behavior.

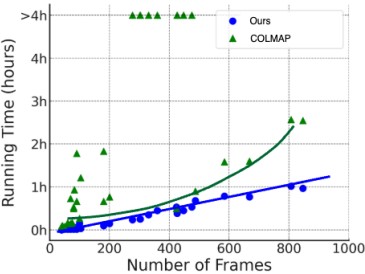

Figure 4: **Runtime Trends.** As the frame count increases, our runtime grows almost linearly, whereas $\text{COL}^{w/omask}$ scales exponentially. The runtime of casualSAM is too large to fit in this figure. The complete runtime is in Table 3, and appendix E.1.1 (table 9, table 10, table 11).

However, the extracted pseudo-supervision always contains moving outliers. To mitigate its impact, without any GT motion priors, we identify such outliers by modeling the uncertainty their presence may cause in the observed distributions of the inlier points. We introduce the uncertainty $\mathbf{\Gamma} \in \mathbb{R}^H$ associated with $\mathbf{P}^{cali} \in \mathbb{R}^H$ and incorporate the Cauchy distribution $f(x; x_0, \mathbf{\Gamma}) = \frac{1}{\pi \mathbf{\Gamma}[1+(\frac{x-x_0}{\mathbf{\Gamma}})^2]}$, $\mathbf{\Gamma} > 0$ to model the uncertainty parameter $\mathbf{\Gamma}$ since this distribution can better handle the heavy tails than, e.g., the Gaussian distribution. As depicted in Figure 3, during optimization, inliers are expected to have low uncertainty, while outliers have high uncertainty. Since the scale parameter $\mathbf{\Gamma}$ in $f(x; x_0, \mathbf{\Gamma})$ is required to be strictly positive, we introduce a new parameter $\mathbf{\Gamma}^{raw}$ which we obtain $\mathbf{\Gamma}$ from using the Softplus function $\mathbf{\Gamma} = \log(1 + e^{\mathbf{\Gamma}^{raw}})$, $\mathbf{\Gamma}^{raw} \in \mathbb{R}^H$. This effectively ensures $\mathbf{\Gamma} \in \mathbb{R}^H_{>0}$ is differentiable and has smooth gradients.

**Losses.** To down-weight outliers by learned $\mathbf{\Gamma}$, we replace the commonly used projection error $\mathbb{E}^{proj} = \|\mathbf{P}^{proj} - \mathbf{P}\|_2^2$ with our proposed **Average Cumulative Projection** (ACP) error, defined as:

$$\mathbb{E}^{ACP}_{h \in [0, H-1]} = \frac{\sum \mathbb{1}_{\{\mathbf{I}=h\}} \circ \|\mathbf{P}^{proj} - \mathbf{P}\|_2^2}{\sum \mathbb{1}_{\{\mathbf{I}=h\}}} \tag{6}$$

, where $\mathbb{E}^{ACP} \in \mathbb{R}^H$ and $\circ$ denotes the element-wise matrix multiplication. For each $\mathbf{P}_h^{cali}$, we accumulate the errors between its corresponding projection and tracking locations across the video, then take the average as $\mathbb{E}_{h \in [0, H-1]}^{ACP}$. Furthermore, we propose the novel Cauchy loss $\mathcal{L}_{cauchy}$ in terms of the negative-log-likelihood $\log\left(\mathbf{\Gamma} + \frac{(x-x_0)^2}{\mathbf{\Gamma}}\right)$ of $f(x; x_0, \mathbf{\Gamma})$ where we replace $x - x_0$ with $\mathbb{E}^{ACP}$ as eq. (7). Our total loss $\mathcal{L}_{total}$ in Equation (8) consists of $\mathcal{L}_{cauthy}$ and a depth regularization term $\mathcal{R}_{depth}$ to encourage positive depth. With the estimated camera parameters, we use 4DGS [47] for scene reconstruction. Reconstruction and loss derivation details are in appendix A and appendix B.

$$\mathcal{L}_{cauchy} = \frac{1}{H} \sum_{h=0}^{H} \log\left(\mathbf{\Gamma} + \frac{(\mathbb{E}^{ACP})^2}{\mathbf{\Gamma}}\right) \tag{7}$$

$$\mathcal{L}_{total} = \mathcal{L}_{cauthy} + \mathcal{R}_{depth}, \ \mathcal{R}_{depth} = \frac{1}{N} \sum_{i=0}^{N} -\text{ReLU}(\mathbf{P}_i^{proj-homo}[:, 3]) \tag{8}$$

### 3.3 Two-stage Optimization Strategy

To avoid convergence to local minima, we propose this strategy based on an analysis of the asymptotic behavior of the Softplus function and the analytical minima of the inner convex term in $\mathcal{L}_{cauthy}$. Stage 1 focuses on rapid convergence, while Stage 2 aims for stable convergence by initializing $\mathbf{\Gamma}^{raw}$ to the ACP error after Stage 1. The effectiveness of it is concretely demonstrated in Table 7.

**Stage 1.** In the Softplus function, $\mathbf{\Gamma} = \log(1 + e^{\mathbf{\Gamma}^{raw}}) \approx \mathbf{\Gamma}^{raw}$, as $\mathbf{\Gamma}^{raw} \to +\infty$. So in Stage 1, we fix $\mathbf{\Gamma}^{raw} = 1$ and optimize only $\mathbf{P}^{cali}$, $f$, $\mathbf{Q}$, and $\mathbf{t}$ for quick convergence. The loss will converge to a certain value beyond the global minimum, as there is no proper $\mathbf{\Gamma}$ to down-weight outliers.

**Stage 2.** The inner term $\Phi = x + \frac{\mathbf{O}}{x}, \mathbf{O} > 0$ of $\mathcal{L}_{cauchy}$ is convex. Assuming a constant $\mathbf{O} \in \mathbb{R}^+$ and solving for $\min_x \Phi(x)$, we have $x^* = \sqrt{\mathbf{O}}$. Similarly, in Stage 2, if $\mathbf{\Gamma}^{raw}$ is randomly initialized to values largely different from $\mathbb{E}_{stage1}^{ACP}$ (the ACP error from Stage 1), convergence will be highly unstable. Therefore, we initialize $\mathbf{\Gamma}^{raw} = \mathbb{E}_{stage1}^{ACP}$, and optimize $\mathbf{P}^{cali}$, $f$, $\mathbf{Q}$, $\mathbf{t}$, and $\mathbf{\Gamma}^{raw}$ jointly.

## 4 Experiments

To demonstrate the superiority of our method, we show extensive quantitative and qualitative results in this section. For NeRF-DS [50], DAVIS [28], and iPhone [7] datasets without GT camera parameters, we feed the camera parameters from different methods to 4DGS [47], while keeping all other factors the same, and evaluate each NVS performance (PSNR, SSIM, and LPIPS). Regarding the MPI-Sintel [2] and TUM-dynamics [36] datasets with GT camera parameters, we directly evaluate methods by ATE, RPE trans, and RPE rot metrics. *In all tables, the best and second-best results are* ***bold*** *and* underline. More about datasets, and evaluation metrics are in appendix C and appendix D.

### 4.1 Implementation Details

The optimization is conducted on 1 NVIDIA A100 40GB GPU with Adam [27] optimizer and learning rates $l_{\mathbf{Q}} = 0.01$, $l_{\mathbf{t}} = 0.01$, $l_f = 1.0$, $l_{\mathbf{P}^{cali}} = 0.01$, and $l_{\mathbf{\Gamma}^{raw}} = 0.01$. We also choose to build our patch-wise tracking filters on CoTracker [13] and load its pre-training weights. The hyperparameters of our patch-wise tracking filters are set at $\tau_{var} = 0.1$, $B = 100$, $w_{\text{NeRF-DS, DAVIS, MPI-Sintel}} = 12$, and $w_{\text{iPhone, TUM}} = 24$. Notably, $w$ is only related to the frame size. Besides, throughout our experiments, we have 200 and 50 iterations in $\text{Stage}_1$ and $\text{Stage}_2$ respectively.

Table 5: **Camera Pose Evaluation on MPI-Sintel [2].** (ATE↓/RPE trans↓/RPE rot↓) We achieve better results than casualSAM [58] and exclude $\text{COL}^{\text{w/o mask}}$ due to its failure.

| Method | alley_1 | alley_2 | ambush_4 | ambush_5 | ambush_6 | market_2 | market_6 |
|---|---|---|---|---|---|---|---|
| casualSAM [58] | 0.028/0.006/0.057 | **0.003**/0.003/0.392 | **0.040**/0.058/**0.321** | **0.053**/0.040/**0.211** | 0.302/**0.088**/2.362 | 0.010/0.010/**0.041** | 0.239/0.207/0.544 |
| Ours | **0.002/0.003/0.038** | 0.009/**0.002/0.047** | 0.119/**0.049**/1.367 | 0.065/**0.039**/1.192 | **0.080**/0.129/**2.191** | **0.003/0.010**/0.110 | **0.009/0.006/0.301** |

| Method | shaman_3 | sleeping_1 | sleeping_2 | temple_2 | mountain_1 | bamboo_1 | bamboo_2 |
|---|---|---|---|---|---|---|---|
| casualSAM [58] | 0.008/0.009/**0.050** | 0.017/0.016/0.173 | 0.013/0.025/0.170 | **0.005**/0.004/0.380 | **0.003**/0.004/0.182 | 0.033/0.009/0.056 | 0.005/0.003/0.035 |
| Ours | **0.003/0.001**/0.085 | **0.008/0.001/0.074** | **0.002/0.001/0.034** | 0.017/**0.003/0.142** | 0.007/**0.004/0.060** | **0.003/0.003/0.033** | **0.004/0.003/0.033** |

Table 2: **NVS Evaluation on NeRF-DS [50] and DAVIS [28].** (PSNR↑/SSIM↑/LPIPS↓) * is supervised by additional GT priors. Ours is the best among these two datasets.

| Method | NeRF-DS | DAVIS |
|---|---|---|
| RoDynRF[20]* | 23.033/0.749/0.385 | - |
| COL^w/ mask * | 32.174/0.923/0.147 | - |
| COL^w/o mask | 29.348/0.875/0.224 | 9.196/0.236/0.435 |
| casualSAM[58] | 21.230/0.686/0.463 | 19.032/0.486/0.482 |
| Ours | **33.552/0.938/0.118** | **22.292/0.709/0.279** |

Table 3: **Runtime Evaluation on NeRF-DS [50], DAVIS [28], and iPhone [7],** covering frame count from 50 to 900. * is supervised by additional GT priors. Our method is the most efficient.

| Method | NeRF-DS | DAVIS | iPhone |
|---|---|---|---|
| RoDynRF [20]* | 29.6h | 27.4h | 28.5h |
| COL^w/ mask * | 1.5h | - | - |
| COL^w/o mask | 1.8h | 0.51h | 9.53h |
| casualSAM [58] | 10.5h | 0.28h | 4.07h |
| Ours | **0.83h** | **0.03h** | **0.33h** |

Table 4: **Camera Pose Evaluation on TUM-dynamics [36].** Other results are from Cut3r [44] and Monst3r [56]. Performance of DROID-SLAM [39] is from casualSAM [58]. Our method achieves the best overall performance among all RGB-only supervised methods, and even better than the ones supervised by additional GT priors.

| Supervision | Method | ATE↓ | RPE trans↓ | RPE rot↓ |
|---|---|---|---|---|
| GT 3D Point Cloud & Camera Pose | Monst3r [56] | 0.098 | 0.019 | 0.935 |
| | Dust3r [46] | 0.083 | 0.017 | 3.567 |
| | Mast3r [18] | **0.038** | **0.012** | 0.448 |
| | Cut3r [44] | 0.046 | 0.015 | 0.473 |
| GT Focal Length + *GT Motion Prior* | LEAP-VO [3] | 0.046 | 0.027 | **0.385** |
| GT Focal Length + *Metric Depth* | DROID-SLAM [39] | 0.043 | - | - |
| GT Motion Priors | ParticleSfM [59] | - | - | - |
| RGB-Only | Robust-CVD [16] | 0.153 | 0.026 | 3.528 |
| | casualSAM [58] | 0.071 | **0.010** | 1.712 |
| | Ours | **0.065** | **0.010** | **0.987** |

## 4.2 Time Efficiency Evaluation

In Table 3, we present the average runtime evaluations. Our average runtime on NeRF-DS, DAVIS, and iPhone is 55%, 11%, and 8% of that of the second-fastest methods, while keeping the best performance as shown in table 2. We attribute it to three main reasons: (1) Our method only leverages the maximally sparse pseudo-supervision extracted by our proposed patch-wise tracking filters under the RGB-only supervision. (2) Our uncertainty parameters are associated with the 3D calibration points rather than 2D uncertainty maps [58], significantly reducing the number of learnable parameters. For `Plate` video (424 frames) in NeRF-DS, casualSAM has ($424 \times 270 \times 480$) uncertainties, whereas our method only has 440 uncertainties, one per $\mathbf{P}^{cali}$. 3) The two-stage optimization strategy highly accelerates the optimization speed. As seen from Table 7, omitting the two-stage strategy leads to a dramatic performance drop after the same iterations, indicating more iterations, and thus time, are needed to achieve the same performance.

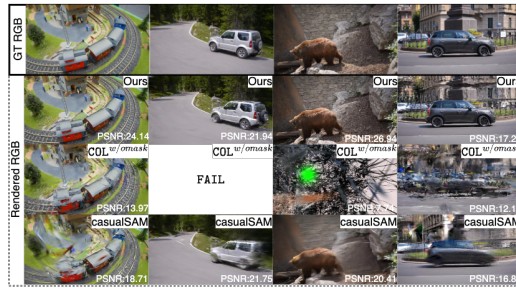

Figure 5: **Qualitative NVS Results on DAVIS [28].** Our performance is the best because of our accurate camera estimates. More are in appendix E.2.2 (fig. 14, fig. 15, fig. 16, and fig. 17).

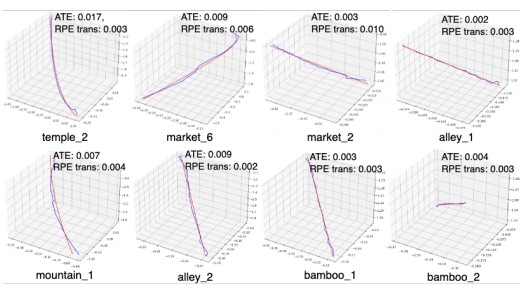

Figure 6: **Qualitative Results of Camera Pose on MPI-Sintel [2].** – represents our camera estimates; – represents the GT. Our estimated camera trajectories almost perfectly align with the GT.

Besides, in Figure 4, we see that our method exhibits a linear growth (at the rate of about 1/800 hours per frame) vs COL^w/omask whose runtime growth is roughly exponential. This difference will be increasingly significant as the video length increases, which can also demonstrate the superior

Table 6: **NVS Evaluation on iPhone [7].** (PSNR↑/SSIM↑/LPIPS↓) `Record3D` is *a paid mobile app* obtaining camera results by LiDAR sensors, where are provided by [7]. Ours is the best among RGB-only supervised methods and surpasses LiDAR-based `Record3D` sometimes.

| Method | Apple | Paper-..mill | Space-out | Backpack | Block | Creeper | Teddy |
|---|---|---|---|---|---|---|---|
| `Record3D` | **26.35/0.77/0.33** | 23.91/0.73/0.24 | 27.12/0.77/0.33 | 20.79/0.56/0.40 | **23.72/0.71/0.38** | 21.80/0.63/0.27 | 19.72/0.59/0.41 |
| COL^w/o mask | 22.45/0.69/0.41 | 22.74/0.72/0.28 | 24.33/0.74/0.38 | 18.58/0.39/0.54 | 18.49/0.60/0.49 | 18.13/0.43/0.48 | 16.56/0.50/0.50 |
| casualSAM[58] | 19.03/0.58/0.57 | 18.85/0.39/0.54 | 22.09/0.67/0.47 | 18.41/0.36/0.55 | 19.10/0.59/0.52 | 16.40/0.29/0.62 | 15.69/0.42/0.58 |
| Ours | 25.96/0.74/0.37 | **24.09/0.74/0.22** | **28.42/0.79/0.31** | **21.22/0.64/0.32** | 23.28/0.69/0.38 | **21.67/0.63/0.28** | **20.78/0.60/0.41** |

| Method | Handwavy | Haru-sit | Mochi-..five | Spin | Sriracha | Pillow | Wheel |
|---|---|---|---|---|---|---|---|
| `Record3D` | 27.80/0.86/0.24 | **29.86/0.87/0.22** | 34.34/0.91/0.24 | 24.85/0.69/0.38 | 31.15/0.87/0.25 | 20.86/0.63/0.41 | 20.78/0.60/0.41 |
| COL^w/o mask | 15.69/0.61/0.55 | 25.58/0.80/0.30 | 22.47/0.77/0.38 | 19.27/0.55/0.49 | 28.41/0.86/0.28 | 14.75/0.46/0.57 | 20.78/0.60/0.41 |
| casualSAM[58] | 20.87/0.68/0.46 | 19.88/0.69/0.41 | 26.34/0.84/0.35 | 19.33/0.45/0.57 | 23.20/0.73/0.42 | 16.95/0.51/0.55 | 14.69/0.47/0.57 |
| Ours | **28.02/0.86/0.22** | 28.31/0.85/0.24 | **34.56/0.92/0.22** | 24.81/0.67/0.39 | **32.49/0.89/0.25** | 20.63/0.61/0.44 | 20.42/0.67/0.37 |

time efficiency of our method compared with other RGB-only supervised methods. We exclude the casualSAM here since its runtime is too large to fit here.

## 4.3 Camera Pose Evaluation

We follow the same evaluation setup of Cut3r [44] and Monst3r [56] on TUM-dynamics [36] and evaluate all videos of the synthetic MPI-Sintel [2] dataset.

**Quantitative Evaluation.** In Table 5 and Table 4, our method has the best performance among all RGB-only supervised approaches. Our method also achieves comparable or even better results than others that require additional GT priors as supervision. We attribute it to our accurate and robust pseudo-supervision, derived from RGB-only input, enabling effective outlier-aware joint optimization. Besides, our uncertainty modeling and loss design effectively down-weight the impact of moving outliers. Since COL^w/ mask and COL^w/o mask always fail on MPI-Sintel [2], as observed by us and [20, 56], we exclude comparisons with them here. However, RGB-only supervised methods including ours perform not very well in some special cases, which is discussed in the limitations.

**Qualitative Evaluation.** In Figure 6, we show our estimated camera trajectories alongside the GT on MPI-Sintel [2]. Our estimated camera trajectories can perfectly overlap with the GT, which provides qualitative support to the higher accuracies seen in the quantitative results in Table 5.

## 4.4 NVS Evaluation

Since NeRF-DS [50], DAVIS [28], and iPhone [7] datasets do not provide GT camera parameters, we follow [6, 20, 45, 19] by inputting camera estimates of different methods into the same 4D reconstruction pipeline - 4DGS [47], and evaluate the NVS performance. Such NVS performance reveals the quality of the camera parameter estimation.

**Quantitative Evaluation.** In Table 2, our method is the best on NeRF-DS [50] (long videos w/ little blur, textureless regions, and specular moving objects) and DAVIS [28] (short videos w/ low parallax and rapid object movement), demonstrating our more accurate camera estimates. We skip COL^w/ mask and RoDynRF on DAVIS [28] because they are not RGB-only supervised methods and require supervision beyond RGB frames, and have already underperformed compared to ours on NeRF-DS [50]. Besides, our pseudo-supervision extraction built on the PT models [13, 12] performs better on low-parallax videos, which remains challenging for the pre-trained depth model [31].

Regarding the iPhone [7] dataset (videos w/ irregular camera movement and object movement), it provides so-called 'GT' camera parameters obtained by `Record3D` which is a paid mobile app obtaining camera results by LiDAR sensors. However, we observe that such so-called 'GT' camera parameters are occasionally unreliable. As shown in Table 6 and fig. 7, besides being the best among all RGB-only supervised methods, our method can occasionally beat `Record3D`.

**Qualitative Evaluation.** We evaluate the quality of the rendered RGB images and depth maps in Figure 7, Figure 8, and Figure 5. Beyond superior RGB renderings, our camera estimates yield the highest-quality depth maps, offering more convincing evidence of accurate scene geometry than RGB renderings. It indicates that our estimated camera parameters enable the model to learn the correct dynamic scene representations rather than overfitting to training views. In the first row of Figure 7, ours performs the best (surpassing even `Record3D`), especially in rendered depth; whereas

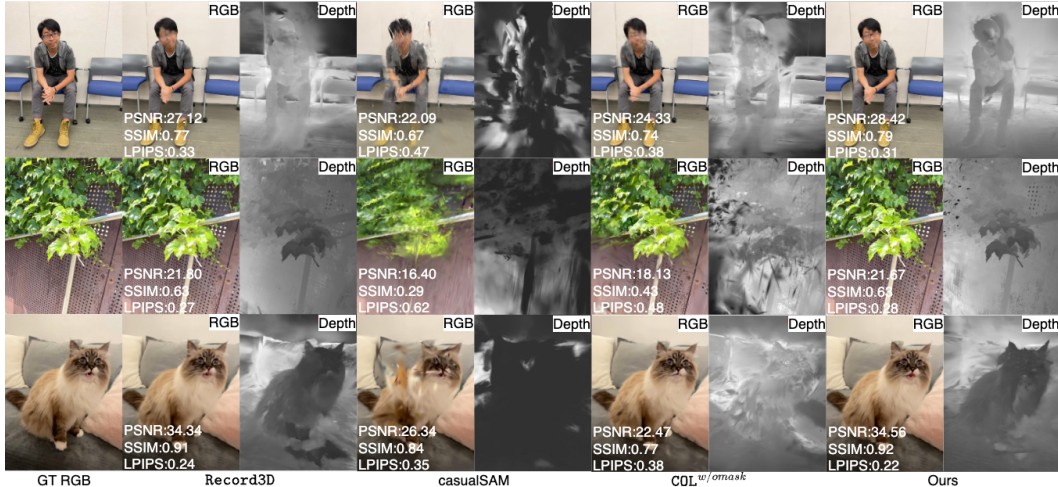

Figure 7: **Qualitative NVS Results on iPhone [7].** Our method outperforms other SOTA RGB-only supervised approaches and even surpasses LiDAR-based `Record3D` when the movement in scenes with large motion (top row). More are in appendix E.2.2 (fig. 11, fig. 12, and fig. 13).

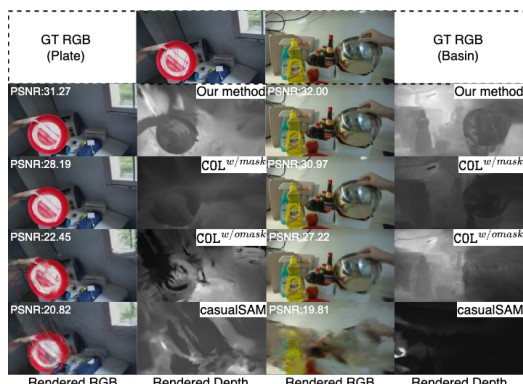

Figure 8: **Qualitative NVS results on NeRF-DS [50].** Our renderings are the most plausible. More are in appendix E.2.2 (fig. 18).

Table 7: **Ablation Study on NeRF-DS [50] - Part 1.** We conduct ablation studies on different scene optimization strategies and individual components of our proposed model.

| Scene Optimization | Camera Optimization | PSNR↑ | SSIM↑ | LPIPS↓ |
|---|---|---|---|---|
| D-3DGS [53] | COL$^{\text{w/o mask}}$ | 31.14 | 0.9192 | 0.1609 |
| | Ours (full) | **32.45** | **0.9336** | **0.1312** |
| 4DGS [47] | COL$^{\text{w/o mask}}$ | 29.35 | 0.8748 | 0.2240 |
| | Ours (full) | **33.55** | **0.9381** | **0.1182** |
| | + *w/o two-stage* | 25.95 | 0.8100 | 0.2668 |
| | + *w/o* $\Gamma$ | 26.44 | 0.8667 | 0.2327 |
| | + *w/o* $\mathbb{E}^{ACP}$ | 23.56 | 0.7203 | 0.3139 |
| | + *w/o texture filt.* | 25.99 | 0.8356 | 0.2536 |
| | + *w/o gradient filt.* | 26.04 | 0.8393 | 0.2404 |
| | + *w/o distrib. filt.* | 26.02 | 0.8382 | 0.2497 |

in the second row, our method does not match `Record3D`, but is still better than other RGB-only supervised works. This is because `Record3D` is not originally designed for dynamic scenes, so when a scene contains larger irregular movements, its performance will be worse (such observations are also supported by the numerical results in Table 6). In contrast, our method is more robust in various scenarios, consistently maintaining high standards.

**Ablation Study.** In Table 7, the loss of any filter results in less robust relations across video, leading to poor camera estimates and NVS performance. Further, the removal of any of $\Gamma$, $\mathbb{E}^{ACP}$, or the two-stage strategy will harm the results due to outliers. This indicates that w/o such a strategy, increasing training iterations is a necessary but not sufficient condition for comparable results.

Table 8: **Ablation Study on NeRF-DS [50] - Part 2.** We conducted ablation studies on different PT models.

| Scene Optimization | Camera Optimization (PT model choice) | PSNR↑ | SSIM↑ | LPIPS↓ |
|---|---|---|---|---|
| 4DGS [47] | Ours + *built on CoTracker [13]* | **33.55** | 0.9381 | 0.1182 |
| | Ours + *built on CoTracker3 [12]* | 33.52 | **0.9384** | **0.1180** |

We also overcome the limitations of COLMAP [32] by improving the performance of different scene optimization models [53, 47] with our camera estimates. As reported in CoTracker3 [12], CoTracker [13] performs worse than CoTracker3. However, in table 8, the performance of our proposed method is nearly independent of building on the particular PT model. This further sup-

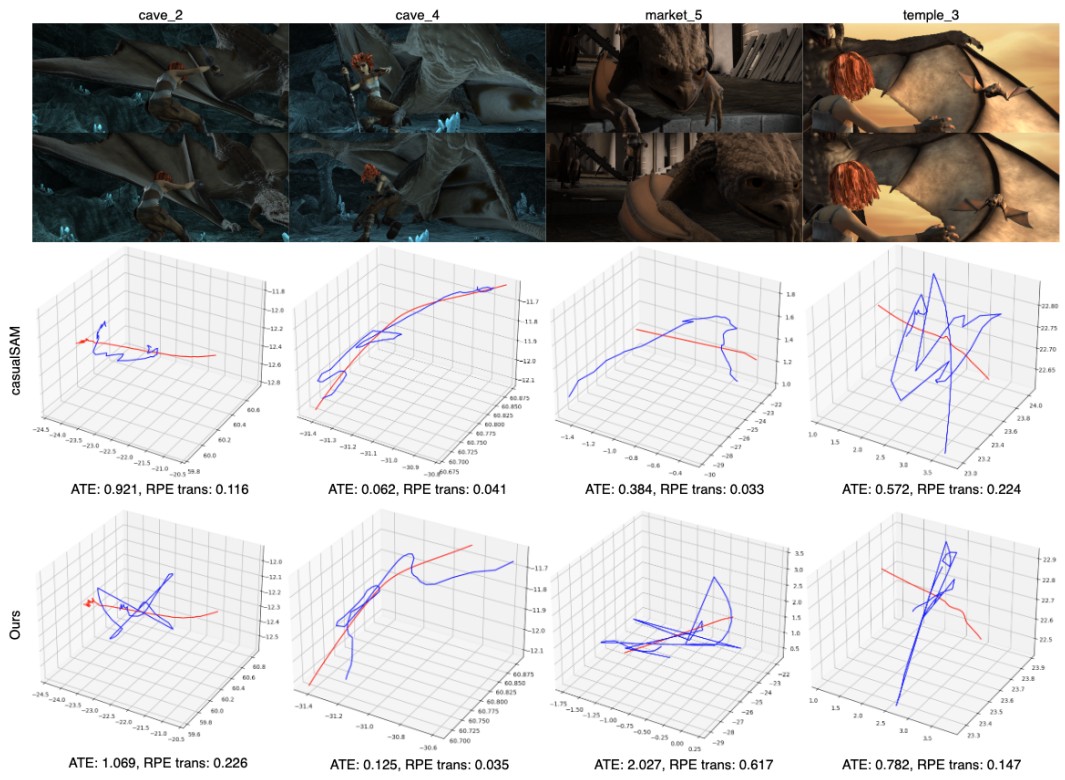

Figure 9: **Failure cases of ours and casualSAM on MPI-Sintel [2].**

ports our claim that our patch-wise tracking filters effectively exact only the accurate trajectories as pseudo-supervision.

## 5 Conclusion and Limitation

We proposed a new RGB-only supervised, accurate, and efficient camera parameter optimization method in casually collected dynamic-scene videos. Our method effectively tackles the challenge of precisely and efficiently estimating per-frame camera parameters under the situation of having no additional GT supervision (e.g. GT motion masks, focal length, 3D point clouds, metric depth, and camera poses) other than RGB videos, which is the most common scenario in real-world and consumer-grade reconstructions. Our method may serve as a step towards high-fidelity dynamic scene reconstruction from casually captured videos.

Although our proposed method is currently the most state-of-the-art RGB-only supervised, accurate, and efficient camera parameter optimization method in dynamic scenes, there are still several limitations. We assume a constant focal length throughout the video. While this assumption is reasonable and currently common to SOTA, the task of accurate and efficient camera parameter optimization for dynamic scene videos with zooming effects under RGB-only supervision remains an open problem. Another common challenge for RGB-only supervised methods, not addressed in this paper, is maintaining robustness in scenes dominated by large moving objects. As shown in fig. 9, the screen space is occupied by the moving human and dragon. It is challenging for our method to establish robust and maximally sparse hinge-like relations as accurate pseudo-supervision because most of the extracted trajectories belong to outliers. CasualSAM [58] struggles due to the rapid changes in depth maps from frame to frame, making 3D space alignment difficult. We plan to maintain consistency in our input setup and address these challenges as part of future research.

## Acknowledgements

The support of the Office of Naval Research under grant N00014-20-1-2444 and of USDA National Institute of Food and Agriculture under grant 2020-67021-32799/1024178 are gratefully acknowledged. This research used the Delta system at NCSA through allocation CIS240124 from the ACCESS program, which is supported by National Science Foundation grants #2138259, #2138286, #2138307, #2137603, and #2138296.

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

## Appendix

Due to the space limit, we show more discussions, results, and details here.

## A Dynamic Scene Optimization

**Preliminary.** As an alternative to NeRF [23], which has a lengthy runtime, 3DGS [14] recently introduced a new way to learn static scene representations in terms of explicit 3D Gaussian ellipsoids. Unlike the implicit representations of NeRF stored as the weights in the Convolutional Neural Network (CNN), 3DGS [14] uses explicit representations in 3D world coordinates and performs differential Gaussian rasterization on GPUs using CUDA which significantly speeds up computational efficiency. Each 3D Gaussian ellipsoid $\mathcal{G}(x)$ is parameterized by its (1) Gaussian center $\mathcal{X} \in \mathbb{R}^3$; (2) quaternion factor $\mathbf{r} \in \mathbb{R}^4$; (3) opacity $\alpha \in \mathbb{R}$; (4) scaling factor $\mathbf{s} \in \mathbb{R}^3$; and (5) color $\mathcal{C} \in \mathbb{R}^k$ (k denotes degrees of freedom), and represented by:

$$\mathcal{G}(x) = e^{-1/2(x-\mu)^T \Sigma^{-1}(x-\mu)} \tag{9}$$

$$\Sigma' = \mathbf{J}\mathbf{W}\Sigma\mathbf{W}^T\mathbf{J}^T, \quad \Sigma = \mathbf{r}\mathbf{s}\mathbf{s}^T\mathbf{r}^T \tag{10}$$

where $\Sigma$ is the 3D covariance matrix in the world space, $\mathbf{W}$ and $\mathbf{J}$ are the view transformation matrix and the Jacobian matrix of the affine transformation parts, respectively, of the projective transformation, and $\Sigma'$ is the covariance matrix in the camera coordinates. The color is rendered by:

$$\mathcal{C}(p) = \sum_{k \in K} c_k \alpha_k \Pi_k^{j-1}(1 - \alpha_k) \tag{11}$$

where $c_k$ and $\alpha_k$ represent the spherical harmonic (SH) coefficient and the density at this point.

**4D GS.** To take advantage of its optimization efficiency, we use 4DGS [47] to learn dynamic scene representations. With different camera estimates and the same scene optimization method [47], we use NVS performance to evaluate the accuracy of camera parameter estimates. In 4DGS[47], the NVS performance depends on how well the canonical representations and deformation representations are optimized. The canonical (refers to 'mean' as in the previous work [14, 53, 47]) representations $\mathcal{G}$ are learned by a canonical Gaussian field to optimize the mean (canonical) position $\mathcal{X} \in \mathbb{R}^3$, color $\mathcal{C} \in \mathbb{R}^k$, opacity $\alpha \in \mathbb{R}$, quaternion factor $r \in \mathbb{R}^4$, scaling factor $s \in \mathbb{R}^3$, and the deformation representations $\mathcal{F}$ are optimized using a deformation field [47] to learn the offsets $\Delta\mathcal{G}$, supervised by an L1 loss between images and renderings. Since the color and opacity of the Gaussian ellipsoids do not change over time, the deformed attributes consist of $(\mathcal{X}', r', s') = (\mathcal{X} + \Delta\mathcal{X}, r + \Delta r, s + \Delta s)$. More details can be found in 4DGS [47].

## B Derivation of Cauchy Negative-log-likelihood

The Cauchy loss function is derived from Equation (12) to Equation (14). Since we use the Cauchy probability density function (PDF) to model the uncertainty of calibration points $\mathbf{P}^{\mathbf{cali}}$, we want to maximize the likelihood of the Cauchy PDF:

$$f(x; x_0, \mathbf{\Gamma}) = \frac{1}{\pi\mathbf{\Gamma}[1 + (\frac{x-x_0}{\mathbf{\Gamma}})^2]}, \quad \mathbf{\Gamma} > 0 \tag{12}$$

Equivalently, we minimize the negative-log-likelihood of $f(x; x_0, \mathbf{\Gamma})$, to define the loss function::

$$\begin{aligned} \text{NLL}(x; x_0, \mathbf{\Gamma}) &= -\log(f(x; x_0, \mathbf{\Gamma})) \\ &= \log(\pi\mathbf{\Gamma}) + \log(1 + (\frac{x-x_0}{\mathbf{\Gamma}})^2) \\ &= \log[\pi \cdot (\mathbf{\Gamma} + \frac{(x-x_0)^2}{\mathbf{\Gamma}})] \\ &= \log\pi + \log(\mathbf{\Gamma} + \frac{(x-x_0)^2}{\mathbf{\Gamma}}) \end{aligned} \tag{13}$$

where $\log \pi$ and $x_0$ denote a constant term and the ground truth which can be omitted. Thus, our objective is as follows:

$$\min_{x,\mathbf{\Gamma}} \text{NLL}(x; x_0, \mathbf{\Gamma}) = \min_{x,\mathbf{\Gamma}} \log(\mathbf{\Gamma} + \frac{(x - x_0)^2}{\mathbf{\Gamma}}) \tag{14}$$

## C   Datasets

To demonstrate our performance on a broader range of scenarios, we have conducted extensive experiments across five public datasets - NeRF-DS [50], DAVIS [28], iPhone [7], MPI-Sintel [2], and TUM-dynamics [36]. These videos contain different camera and object motion patterns, and different texture levels. The lengths of the videos range from about 50 to 900. Regarding the train/test split of the NVS evaluation, for every 2 adjacent frames, we take the first frame for training and the second frame for testing. For the setup of camera pose evaluation, we follow Cut3r [44] and Monst3r [56] in the experiments on TUM-dynamics and evaluate all videos in MPI-Sintel [2].

**NeRF-DS.** NeRF-DS [50] dataset includes seven long monocular videos (400-800 frames) of different dynamic, real-world indoor scenarios with little blur. Each video has at least one specular moving object against a mix of low-texture and high-texture backgrounds. NeRF-DS [50] exhibits large scene and camera movements, so the frames have some blur. The GT motion masks provided are human-labeled and the camera parameters are estimated by $\text{COL}^{\text{w/ mask}}$. Like previous works [50, 53], we take the highest resolution images available ($480 \times 270$) as the RGB input in all experiments.

**DAVIS.** DAVIS [28] dataset contains 40 short monocular videos that capture different dynamic scenes in the wild. Each video has 50-100 frames, including at least one dynamic object. The GT motion masks are also provided as in NeRF-DS [50]. However, like [20, 45, 58, 59], we exclude some videos using fixed cameras, changeable focal lengths, etc. Different from others [20, 45, 58, 3] which only show experiments of about 10 videos in DAVIS [28], we conduct experiments on 21 videos containing large camera and object movements. We utilize the RGB frames with the resolution of $854 \times 480$ as input.

**iPhone.** The iPhone [7] dataset is an extremely challenging dataset (180-475 frames) with significant camera rotations and translations, and rapid movements of objects. There are 14 monocular videos including indoor and outdoor scenes and no GT motion mask is provided. It would also be difficult to insert motion masks for this dataset because there is no clear boundary between the moving and stable regions within any frame. They represent real-world casually recorded videos. We conduct experiments on all of them. The frame size is $720 \times 960$. These videos are recorded by the Record3D [33] app on iPhone which uses LiDAR sensors to obtain metric depth for camera estimation. In our comparisons with the camera estimates provided by Record3D, we also take the Record3D app as one of the baselines and compare with it.

**MPI-Sintel.** MPI-Sintel [2] is a synthetic dataset provided GT camera parameters. It has 18 short videos (about 50 frames) in total containing large object movement. In some cases, the moving objects cover most of the screen. Most of the existing works [20, 56, 3] select 14 videos for evaluation, but in this paper, we evaluate the methods among all the videos. The synthetic MPI-Sintel dataset exhibits domain gaps compared to real-world scenarios, which is considered to be one of the reasons why some existing methods [58, 59] perform well on MPI-Sintel, but do not work efficiently on other real-world datasets. In experiments, we take the frames with default sizes as the input the our method, while keeping the default resizing setup of the other methods.

**TUM-dynamics.** TUM-dynamics [36] dataset contains 8 long real-world blurry videos recording the dynamic indoor scenes provided with GT camera parameters. However, although the videos in this dataset are indoor scenes, each video features a significant depth of field. TUM-dynamics dataset also contains large camera movement and rapid object movement. We follow the experimental setup of MonST3R [56] on this dataset, which is sampling the first 90 frames with the temporal stride of 3 to save compute.

## D   Evaluation Metrics

As discussed in section 4 of the main paper, we directly conduct camera pose evaluation against the GT on MPI-Sintel[2] and TUM-dynamics [36], using the standard metrics: ATE, RPE trans, and RPE rot. Besides, regarding NeRF-DS [50], DAVIS [28], and iPhone [7] datasets which are not provided

with GT camera parameters, we conduct NVS evaluation with standard metrics: PSNR, SSIM, and LPIPS. We also employ time evaluation to demonstrate the superior time efficiency of our method.

## D.1 PSNR & SSIM & LPIPS

**PSNR.** PSNR is a measure of the ratio between the maximum possible power of a signal and the power of corrupting noise that affects the fidelity of its representation. PSNR is commonly used to compare the qualities of the original and the rendered images, and is obtained from the Mean Square Error (MSE) between the original and the rendered images:

$$\text{PNSR} = 10 \cdot \log_{10}\left(\frac{\text{MAX}^2}{\text{MSE}(\text{Image}_{\text{Rendered}}, \text{Image}_{\text{GT}})}\right) \tag{15}$$

, where MAX is the maximum pixel value of the image.

**SSIM.** SSIM measures the similarity between two images based on structural information. Its evaluation involves luminance, contrast, and structure. Compared to PSNR, SSIM is intended to match human perception more closely. The SSIM values range from -1 to 1, where 1 denotes perfect. It is given as:

$$\text{SSIM} = \frac{(2\mu_x\mu_y + c_1)(2\sigma_{xy} + c_2)}{(\mu_x^2 + \mu_y^2 + c_1)(\sigma_x^2 + \sigma_y^2 + c_2)} \tag{16}$$

where $x$ and $y$ are two images, $\mu_x$, $\mu_y$ and $\sigma_x^2$, $\sigma_y^2$ are the corresponding averages and variances of $x$ and $y$, $\sigma_{xy}$ represents the covariance of $x$ and $y$, and $c_1$ and $c_2$ denote the regularization terms.

**LPIPS.** LPIPS measures perceptual similarity in terms of features of deep neural networks, such as pre-trained VGG [34] or AlexNet [17]. It compares feature activations of image patches. Like [14, 53, 47, 19], we here use the VGG-based networks.

## D.2 ATE & RPE trans & RPE rot

**ATE.** ATE quantifies the difference between the actual trajectory and the estimated trajectory of a robot or camera over time, offering a global measure of error along the entire path. It is calculated by aligning the estimated trajectory with the ground truth and then measuring the Euclidean distance between each corresponding point on the two trajectories.

**RPE trans.** RPE trans quantifies the error in the translational component between consecutive poses or over a fixed time/distance interval. Unlike ATE, which assesses the overall trajectory, RPE Trans emphasizes the local accuracy of the motion estimation by evaluating how well the system preserves the relative motion between two points in time or space.

**RPE rot.** RPE rot quantifies the error in the orientation component between the estimated poses and the ground truth. This metric is computed by measuring the difference in orientation over short sequences, and it is typically expressed in angular units, such as degrees or radians.

# E   More Results

## E.1   Quantitative Results

### E.1.1   Runtime

We report the detailed runtime comparisons on the NeRF-DS [50], DAVIS [28], and iPhone [7] datasets, each containing over 50 frames, where runtime differences become more pronounced. Specifically, in Table 9 and Table 10, the runtime of our method is the shortest. In addition, in Table 11, on the NeRF-DS [50] dataset, the runtime of $\text{COL}^{\text{w/ mask}}$ and $\text{COL}^{\text{w/o mask}}$ on `Plate` video is shorter than of ours. This is because $\text{COL}^{\text{w/ mask}}$ and $\text{COL}^{\text{w/o mask}}$ fail on this video, leading to a quick convergence to the local minima. Such a conclusion can also supported by qualitative results of fig. 8 in the main paper.

Table 9: **Quantitative Runtime Results on DAVIS [28].** `Cam` → camera optimization time; `Cam+Scene` → overall (camera+scene) optimization time; h → hour; m → minute. We mark the shortest time in **bold**. Our method is the most efficient without any failure.

| Method | Ours | | COL^(w/o mask) | | casualSAM | |
| --- | --- | --- | --- | --- | --- | --- |
| | Cam | Cam+Scene | Cam | Cam+Scene | Cam | Cam+Scene |
| Camel | **1.57m** | **25.28m** | 40m | 71m | 24m | 46m |
| Bear | **3.15m** | **1.08h** | 56m | 88m | 20m | 39m |
| Breakdance-flare | **1.73m** | **0.92h** | FAIL | - | 15m | 34m |
| Car-roundabout | **4.97m** | **21.95m** | 10m | 41m | 18m | 46m |
| Car-shadow | **0.93m** | **17.88m** | 5m | 31m | 10m | 36m |
| Car-turn | **2.97m** | **17.35m** | FAIL | - | 27m | 49m |
| Cows | **2.85m** | **22.50m** | 73m | 84m | 26m | 48m |
| Dog | **1.60m** | **11.70m** | 10m | 53m | 12m | 31m |
| Dog-agility | **0.67m** | **26.63m** | FAIL | - | 5m | 31m |
| Goat | **2.10m** | **18.95m** | 107m | 124m | 22m | 44m |
| Hike | **4.20m** | **31.83m** | FAIL | - | 20m | 42m |
| Horsejump-high | **1.97m** | **21.20m** | 5m | 33m | 10m | 31m |
| Lucia | **1.97m** | **22.75m** | 44m | 65m | 16m | 36m |
| Motorbike | **2.08m** | **18.77m** | 6m | 31m | 9m | 32m |
| Parkour | **9.07m** | **26.65m** | 16m | 37m | 27m | 48m |
| Rollerblade | **1.25m** | **17.08m** | FAIL | - | 8m | 27m |
| Tennis | **3.47m** | **17.03m** | 9m | 30m | 17m | 38m |
| Train | **1.90m** | **18.22m** | 32m | 57m | 19m | 44m |
| Mean | **2.68m** | **21.02m** | 31m | 56m | 17m | 39m |

Table 10: **Quantitative Runtime Results on iPhone [7].** `Cam` → camera optimization time; `Cam+Scene` → overall (camera+scene) optimization time; h → hour; m → minute. We mark the shortest time in **bold**. Our method is the most efficient.

| Method | Ours | | COL^(w/o mask) | | casualSAM | |
| --- | --- | --- | --- | --- | --- | --- |
| | Cam | Cam+Scene | Cam | Cam+Scene | Cam | Cam+Scene |
| Apple | **33m** | **47m** | 10.95h | 11.25h | 6.80h | 7.32h |
| Paper-windmill | **15m** | **27m** | 8.18h | 8.70h | 3.50h | 3.97h |
| Space-out | **23m** | **69m** | 4.42h | 4.72h | 5.87h | 6.30h |
| Backpack | **7m** | **25m** | 1.83h | 2.12h | 1.58h | 2h |
| Block | **27m** | **42m** | 10h | 10.45h | 6h | 6.50h |
| Creeper | **27m** | **45m** | 16.03h | 16.45h | 4.38h | 4.82h |
| Handwavy | **15m** | **30m** | 4.62h | 5.1h | 3.30h | 3.72h |
| Haru-sit | **10m** | **25m** | 0.77h | 1.18h | 1.25h | 1.72h |
| Mochi-high-five | **6m** | **18m** | 0.67h | 1.03h | 1.53h | 1.93h |
| Pillow | **21m** | **36m** | 19.18h | 19.70h | 3.70h | 4.20h |
| Spin | **30m** | **44m** | 20h | 20.58h | 5.50h | 6h |
| Sriracha-tree | **15m** | **27m** | 4.58h | 4.88h | 3.22h | 3.68h |
| Teddy | **31m** | **46m** | 19h | 19.71h | 6.78h | 7.28h |
| Wheel | **27m** | **46m** | 6m | 13.80h | 3.67h | 4.17h |
| Mean | **20m** | **38m** | 9.53h | 9.97h | 4.07h | 4.53h |

Table 11: **Quantitative runtime results on NeRF-DS [50].** Cam $\rightarrow$ camera optimization time; Cam+Scene $\rightarrow$ overall (camera+scene) optimization time; h $\rightarrow$ hour; m $\rightarrow$ minute. We mark the shortest time in **bold**. we show only Cam+Scene of RoDynRF [20] due to its joint optimization of the camera and scene. $*$ is supervised by additional GT priors. COL$^{w/mask}$ and COL$^{w/omask}$ are faster than us on Plate because they fail on this video, leading to a quick convergence to the local minima, which can be supported by qualitative results of fig. 8 in the main paper. Among all, our method is the most efficient.

| Method | Ours | | COL$^{w/ mask *}$ | | COL$^{w/o mask}$ | | casualSAM | | RoDynRF$^*$ |
|---|---|---|---|---|---|---|---|---|---|
| | Cam | Cam+Scene | Cam | Cam+Scene | Cam | Cam+Scene | Cam | Cam+Scene | Cam+Scene |
| Bell | **1.05h** | **1.20h** | 2.50h | 2.72h | 3.00h | 3.25h | 16.5h | 16.8h | 28.6h |
| As | **0.95h** | **1.08h** | 2.00h | 2.17h | 2.55h | 2.72h | 14.87h | 15.08h | 33.6h |
| Basin | **0.75h** | **0.92h** | 1.42h | 1.62h | 1.60h | 1.85h | 9.88h | 10.67h | 33.8h |
| Plate | 0.53h | 0.68h | **0.42h** | **0.60h** | 0.50h | 0.87h | 4.67h | 4.98h | 25.6h |
| Press | **0.68h** | **0.82h** | 0.85h | 1.05h | 0.90h | 1.08h | 6.28h | 6.60h | 28.5h |
| Cup | **1.02h** | **1.15h** | 2.37h | 2.58h | 2.57h | 2.73h | 13.50h | 13.78h | 28.8h |
| Sieve | **0.78h** | **0.92h** | 1.15h | 1.35h | 1.58h | 1.77h | 7.83h | 8.13h | 28.3h |
| Mean | **0.83h** | **0.97h** | 1.52h | 1.73h | 1.82h | 2.03h | 10.50h | 10.80h | 29.6h |

## E.2 Qualitative Results

### E.2.1 Trajectories from Patch-wise Tracking Filters as Pseudo-supervision

In Figure 10, we show the trajectory comparisons on the NeRF-DS [50] dataset as samples. As discussed in the 3rd paragraph in section 1, and section 3.1 of the main paper, our patch-wise tracking filters can establish robust and maximally sparse hinge-like relations as accurate pseudo-supervision, avoiding noisy and inaccurate tracking trajectories. In Figure 10, it is easy to see our proposed method avoids the inaccurate ones in the low-texture regions (walls), and meanwhile, adaptively adds new reliable trajectories when the number of left trajectories on each frame is less than $B$.

### E.2.2 NVS

We show more RGB and depth rendering results on NeRF-DS [50], DAVIS [28], and iPhone [7] dataset in Figure 11, Figure 12, Figure 13, Figure 14, Figure 15, Figure 16, Figure 17, and Figure 18. It is easy to see that the RGB and depth rendering results of our method are better than other RGB-only supervised approaches. In addition, the performance of our method is also comparable with that of the LiDAR-based Record3D app.

### E.2.3 Optimized 3D Gaussian Fields

Since the iPhone [7] dataset is the most challenging dataset with large camera and object movements, we show more visualizations of optimized 3D Gaussian fields in Figure 20, Figure 21, and Figure 19. Such comparisons demonstrate that our camera estimates enable superior reconstruction of 3D Gaussian fields compared to other RGB-only supervised approaches. Moreover, the reconstructed fields using our estimates are comparable to, or even surpass, those obtained with the LiDAR-based Record3D [33] app, particularly in scenes with significant motion.

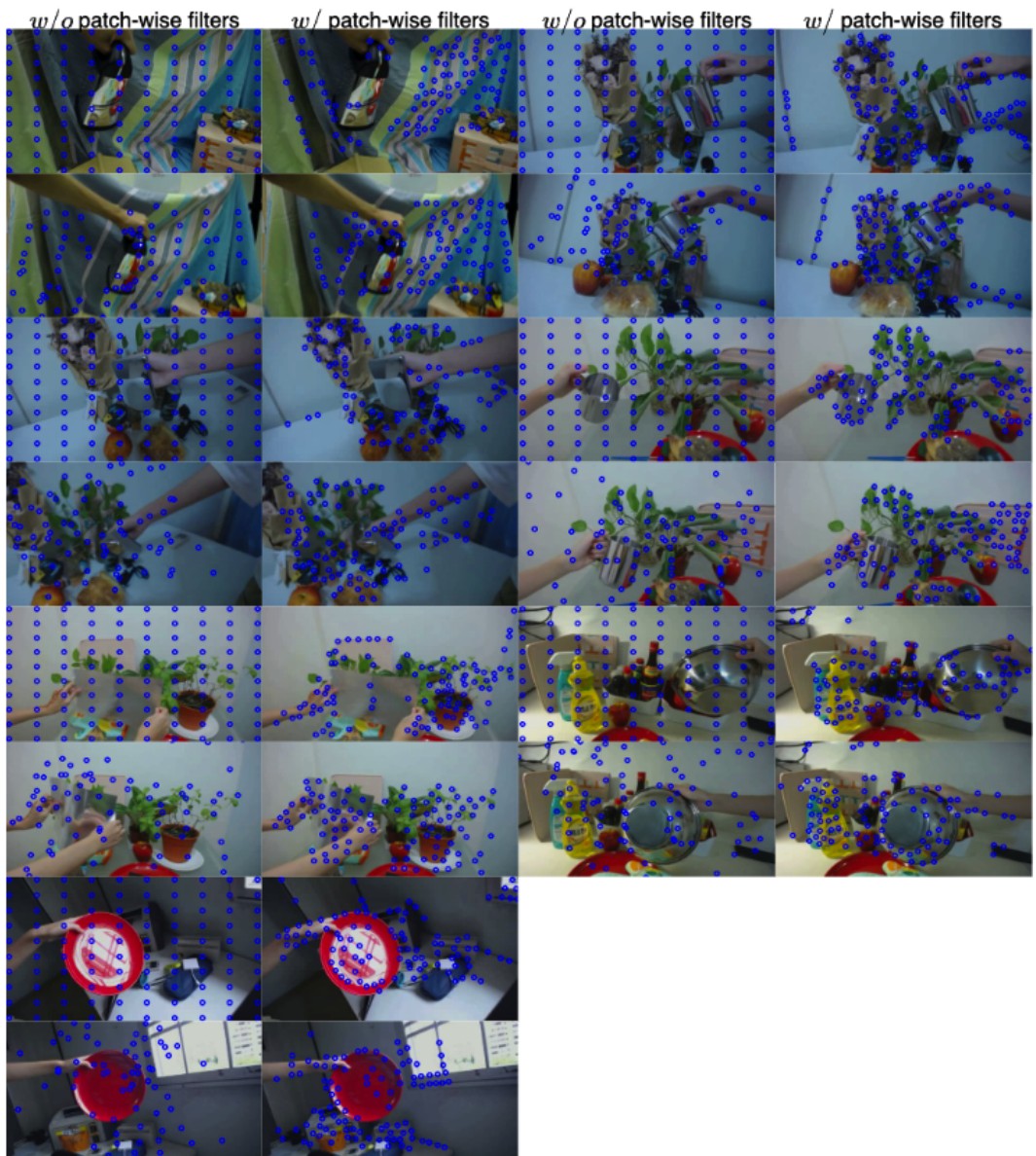

Figure 10: **Trajectory Comparisons on the NeRF-DS [50] Dataset.** In each scenario, top row $\rightarrow$ $F_0$; bottom row $\rightarrow F_{247}$; $w/o$ patch-wise filters $\rightarrow$ raw CoTracker [13]; $w/$ patch-wise filters $\rightarrow$ Ours. It is easy to see our proposed method avoids the inaccurate trajectories in the low-texture regions, whereas the trajectories of the points in the low-texture regions tracked by raw CoTracker [13] are extremely unreliable.

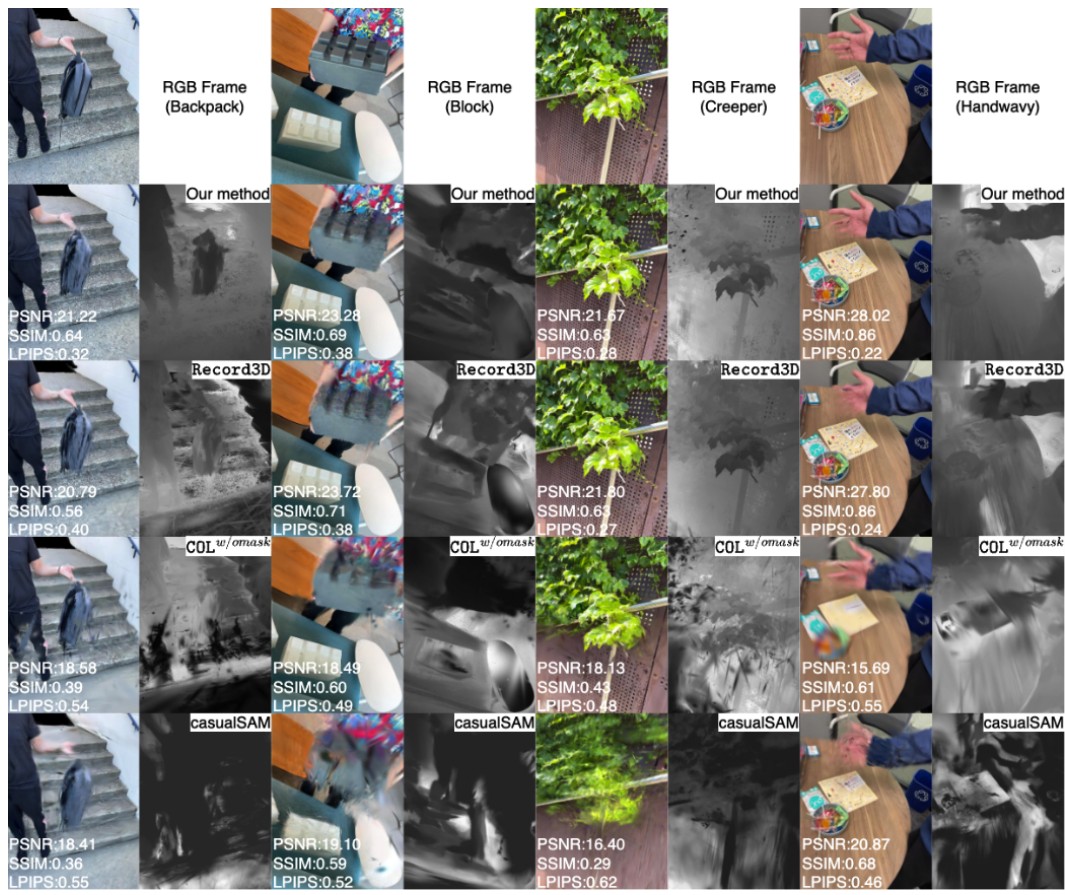

Figure 11: **More Qualitative NVS Results on iPhone [7] - Part 1.** Our renderings exhibit higher fidelity and more accurate geometry compared to other RGB-only supervised methods. Besides, our performance is comparable with, or even better than, the ones of the LiDAR-based Record3D app.

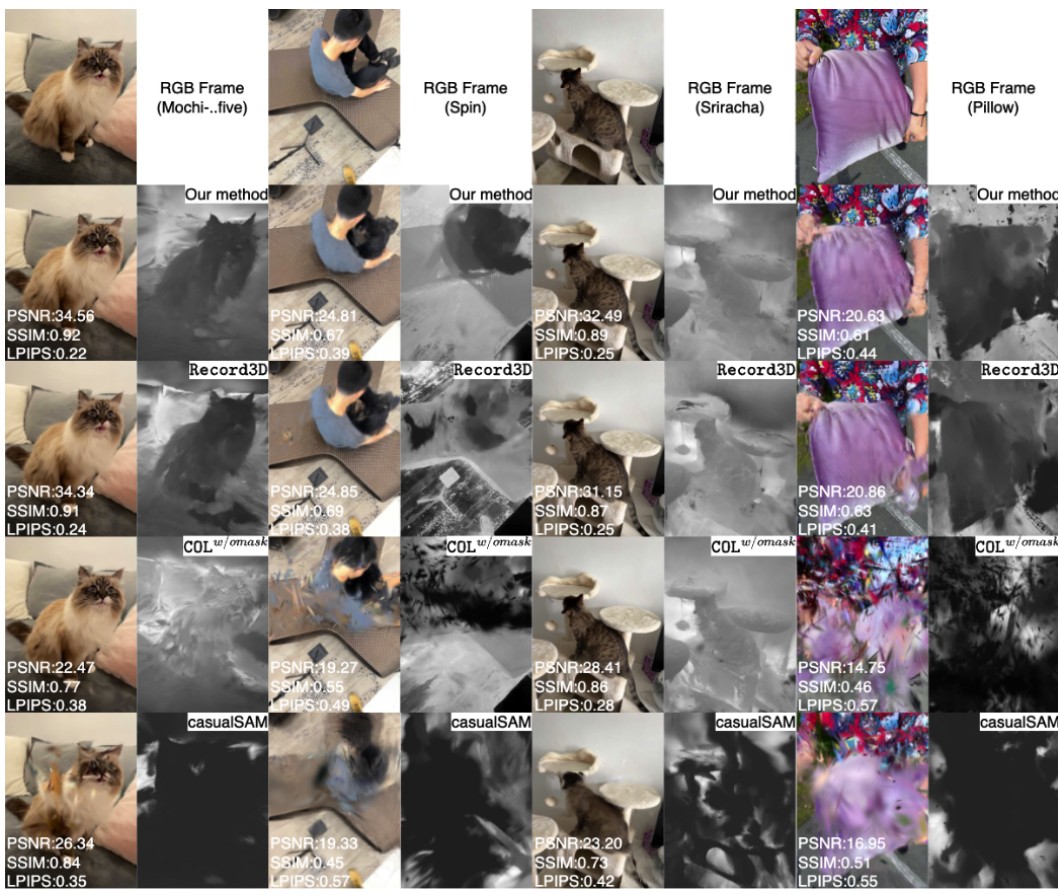

Figure 12: **More Qualitative NVS Results on iPhone [7] - Part 2.** Our renderings exhibit higher fidelity and more accurate geometry compared to other RGB-only supervised methods. Besides, our performance is comparable with, or even better than, the ones of the LiDAR-based Record3D app.

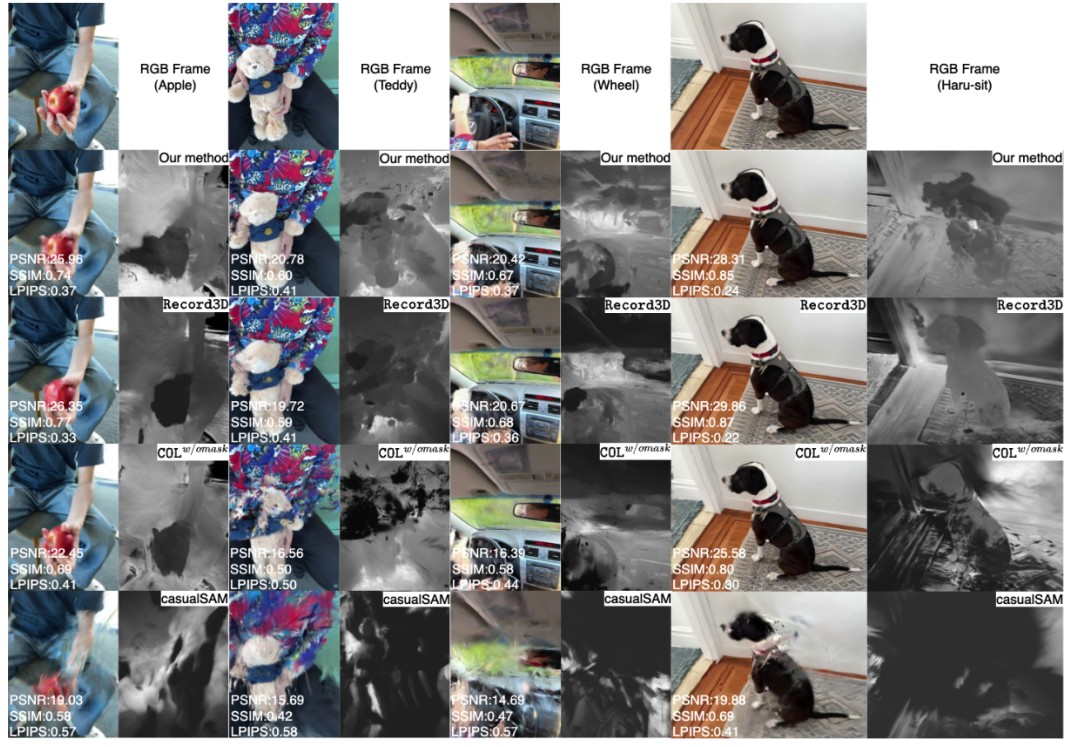

Figure 13: **More Qualitative NVS Results on iPhone [7] - Part 3.** Our renderings exhibit higher fidelity and more accurate geometry compared to other RGB-only supervised methods. Besides, our performance is comparable with, or even better than, the ones of the LiDAR-based Record3D app.

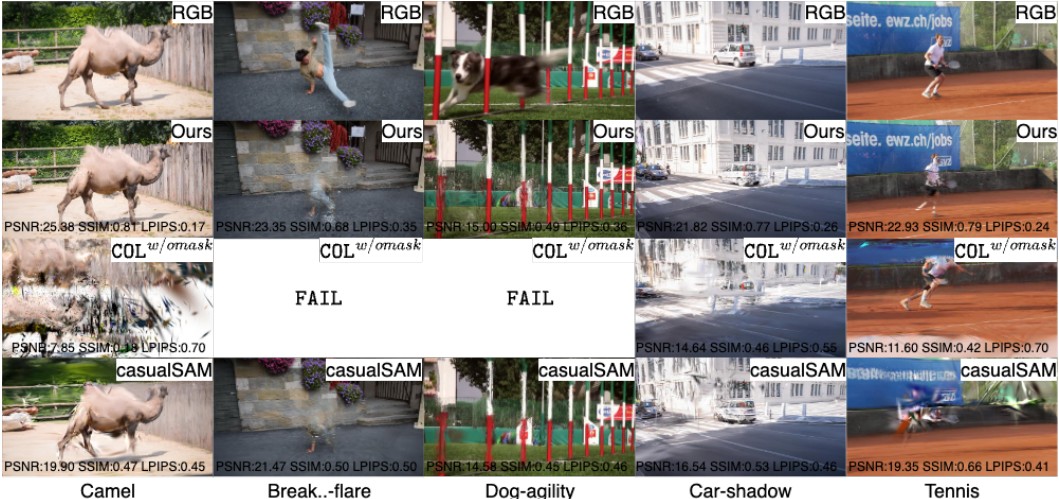

Figure 14: **More Qualitative NVS Results on DAVIS [28] - Part 1.** Our renderings exhibit higher fidelity compared to other RGB-only supervised methods.

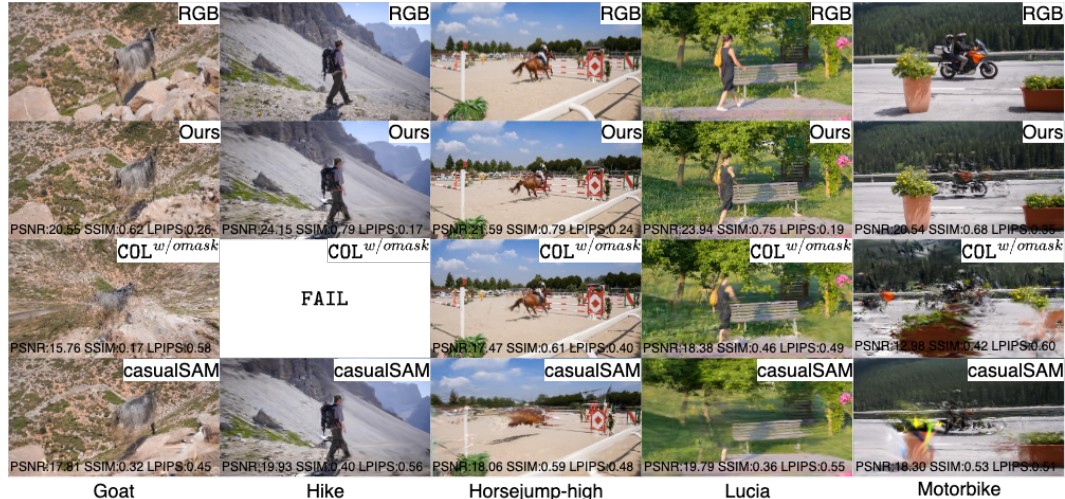

Figure 15: **More Qualitative NVS Results on DAVIS [28] - Part 2.** Our renderings exhibit higher fidelity compared to other RGB-only supervised methods.

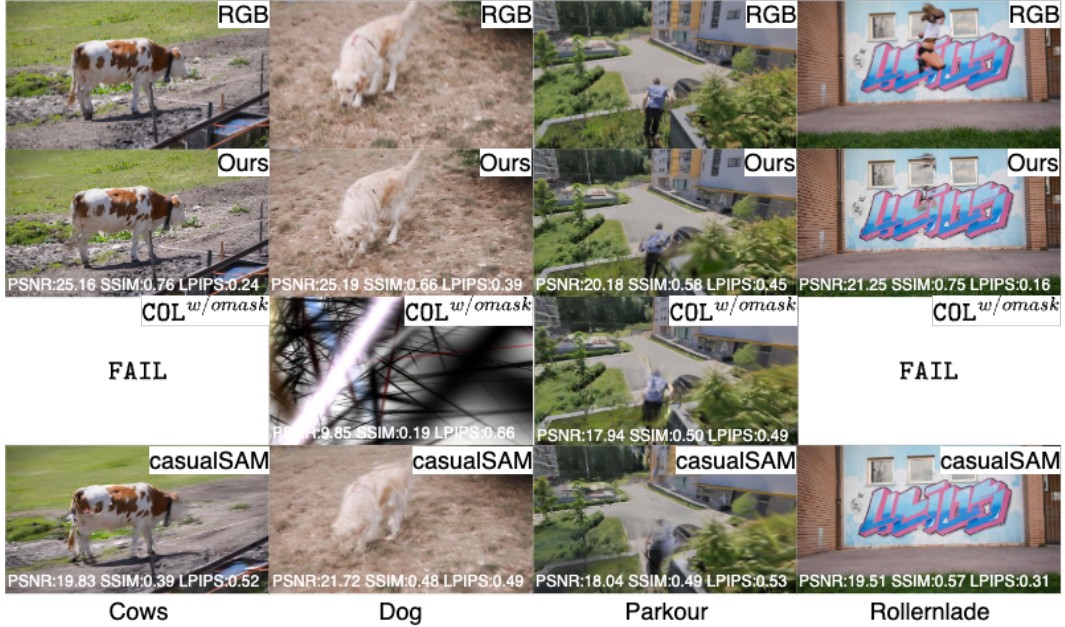

Figure 16: **More Qualitative NVS Results on DAVIS [28] - Part 3.** Our renderings exhibit higher fidelity compared to other RGB-only supervised methods.

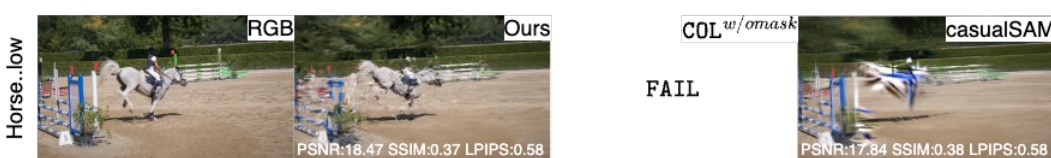

Figure 17: **More Qualitative NVS Results on DAVIS [28] - Part 4.** Our renderings exhibit higher fidelity compared to other RGB-only supervised methods.

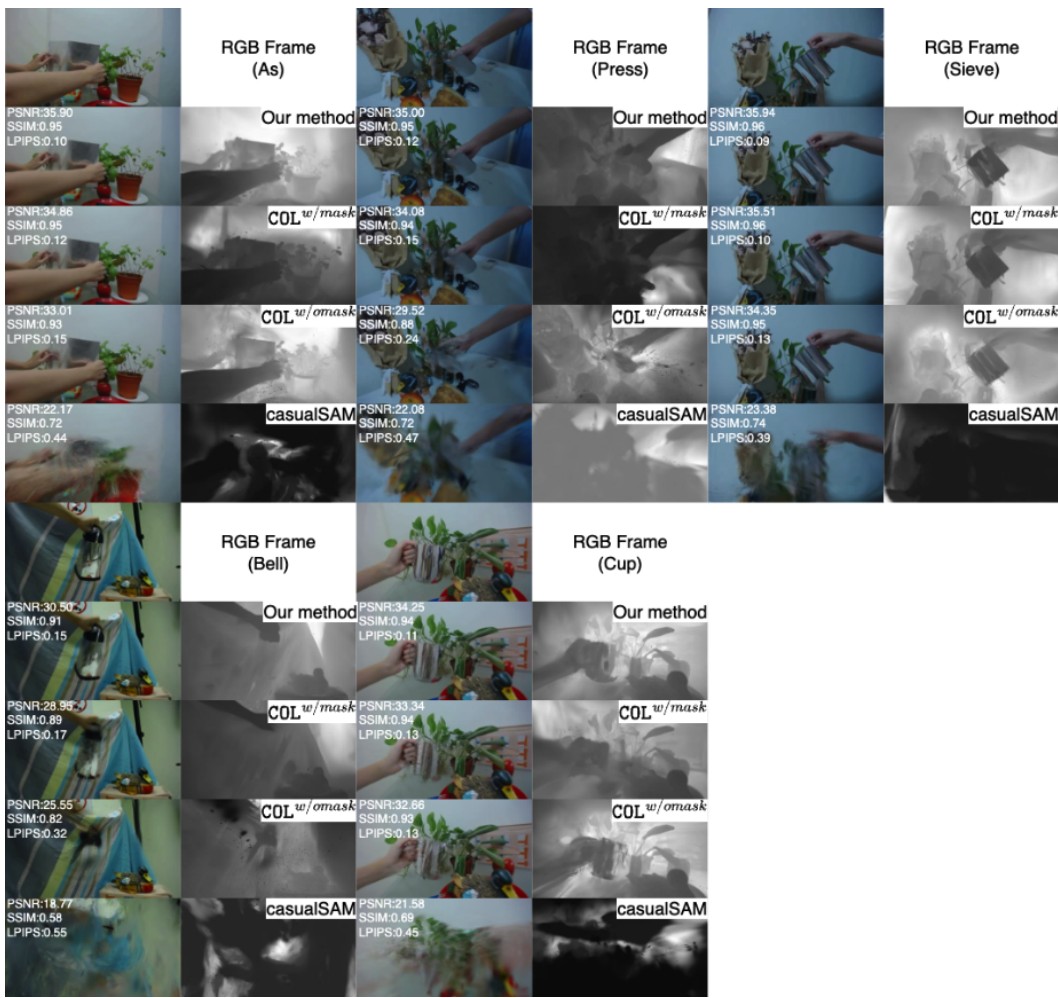

Figure 18: **More Qualitative NVS Results on NeRF-DS [50].** Our renderings exhibit higher fidelity compared to other RGB-only supervised methods.

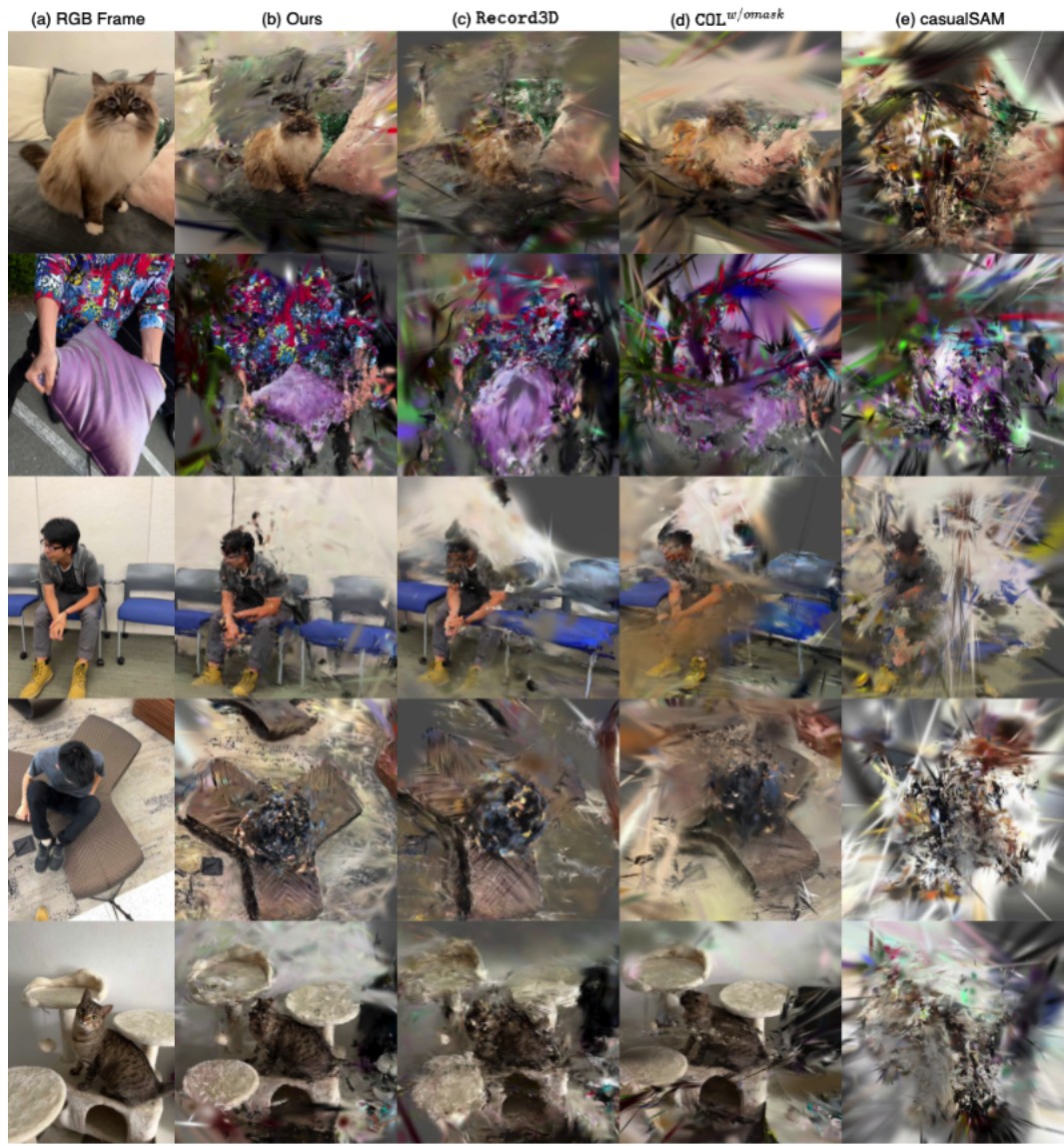

Figure 19: **Optimized 3D Gaussian Fields on iPhone [7] - Part 1.** Our reconstructed 3D Gaussian Fields are more geometrically accurate compared to the ones of other RGB-only supervised methods, which demonstrates our camera estimates are more accurate. Besides, our performance is comparable with, or even better than, the ones of the LiDAR-based Record3D app.

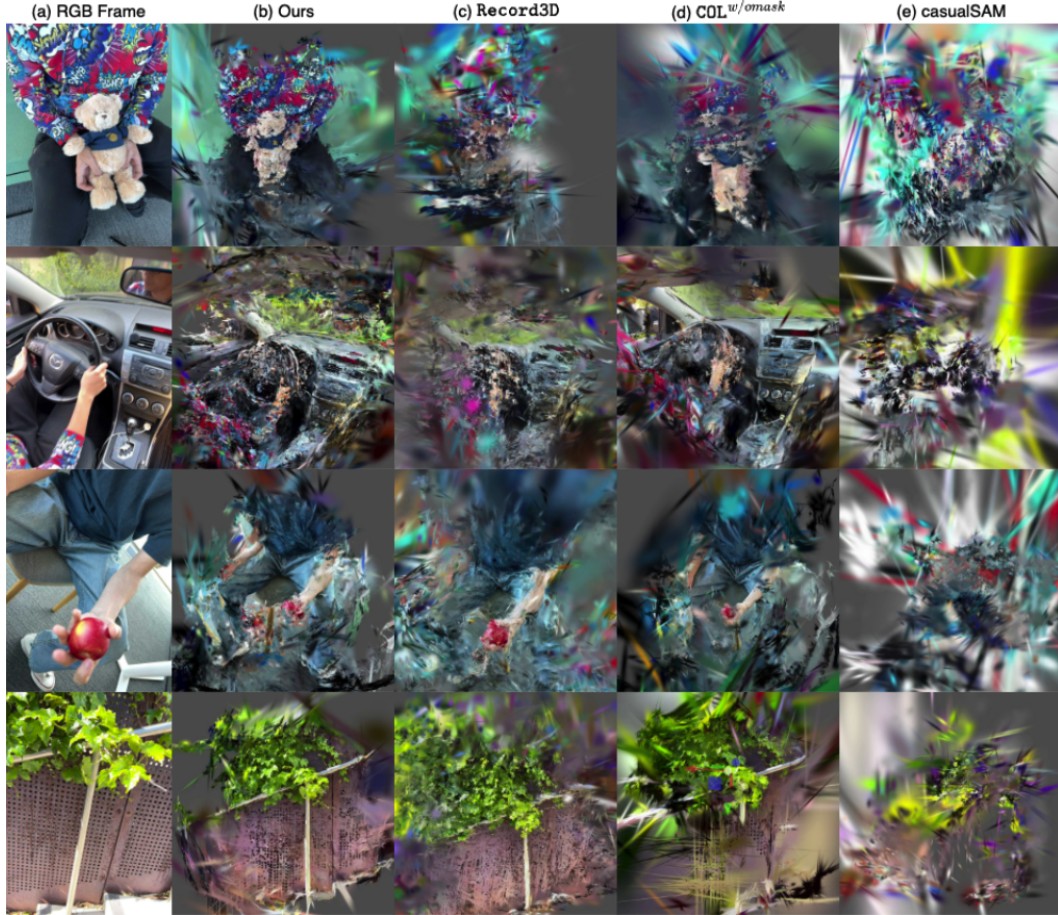

Figure 20: **Optimized 3D Gaussian Fields on iPhone [7] - Part 2.** Our reconstructed 3D Gaussian Fields are more geometrically accurate compared to the ones of other RGB-only supervised methods, which demonstrates our camera estimates are more accurate. Besides, our performance is comparable with, or even better than, the ones of the LiDAR-based Record3D app.

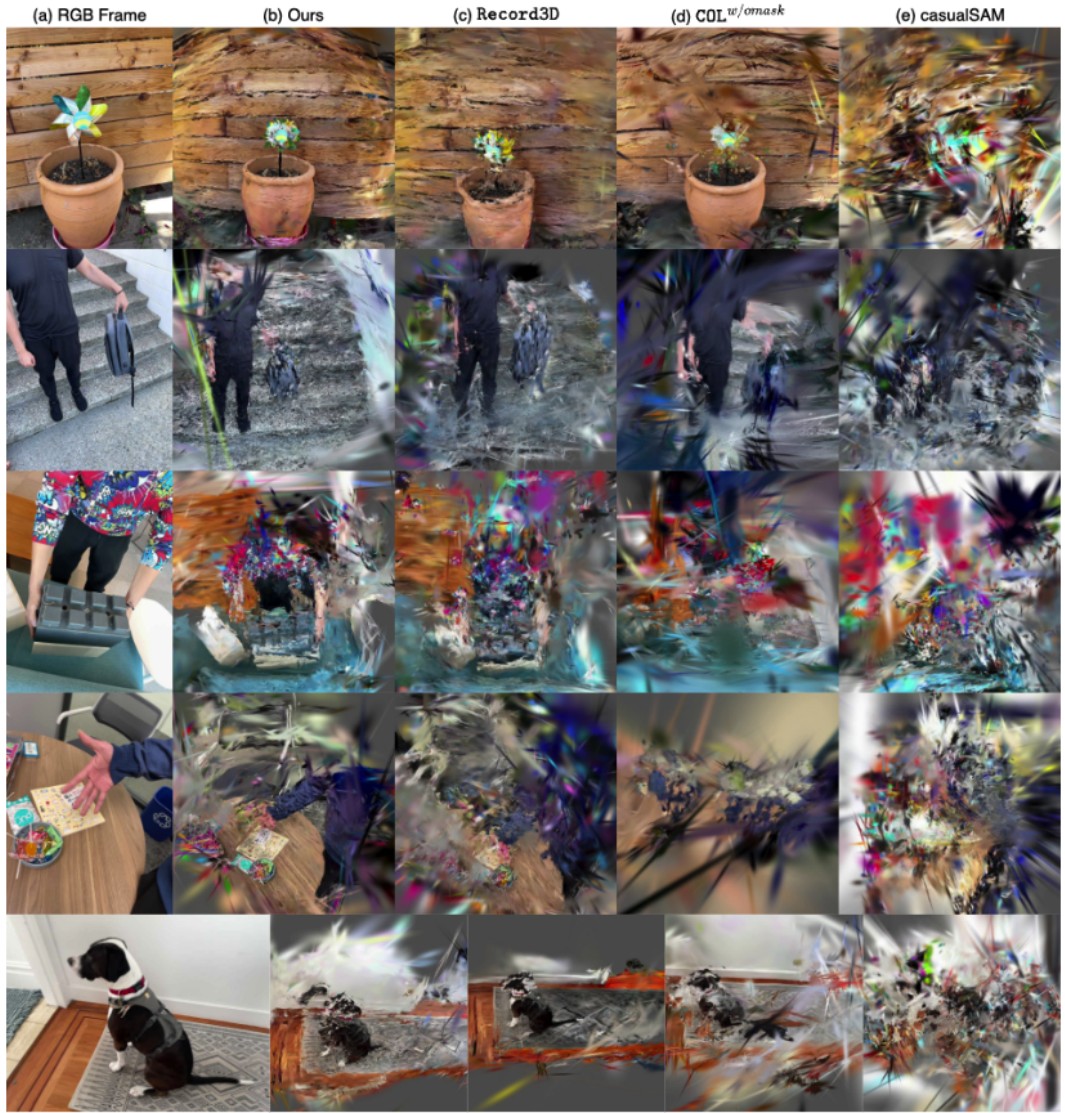

Figure 21: **Optimized 3D Gaussian Fields on iPhone [7] - Part 3.** Our reconstructed 3D Gaussian Fields are more geometrically accurate compared to the ones of other RGB-only supervised methods, which demonstrates our camera estimates are more accurate. Besides, our performance is comparable with, or even better than, the ones of the LiDAR-based Record3D app.

