# OpenReview forum: "RGB-Only Supervised Camera Parameter Optimization in Dynamic Scenes"
_NeurIPS.cc/2025/Conference — NeurIPS 2025 spotlight_

### Official Review · Reviewer_9gnd · 2025-06-17

**Clarity:** 2
**Significance:** 4
**Originality:** 4
**Rating:** 5
**Confidence:** 4

**Summary:**

This paper addresses the challenging task of estimating camera parameters in dynamic scenes. Unlike previous methods that rely on ground-truth labels or require ground-truth masks of moving objects, the proposed approach is built entirely on a point tracking model and performs joint optimization based solely on the extracted point tracks. From my perspective, this method can be viewed as a form of bundle adjustment that is robust to dynamic objects. The results are impressive on real-world videos, even surpassing some supervised methods and commercial software such as that used on iPhones.

**Questions:**

Do other point trackers work as well?

**Ethical Concerns:**

["NO or VERY MINOR ethics concerns only"]

**Final Justification:**

My concerns have been addressed in the rebuttal, and I have decided to maintain my original rating.

**Quality:**

3

**Strengths And Weaknesses:**

Strengths

1. The results are impressive—the proposed method outperforms many supervised approaches on real-world videos, despite relying solely on RGB inputs without any ground-truth supervision.

2. The paper provides a clear and structured comparison with existing methods, particularly by categorizing them based on the type of supervision they use. This helps position the contribution of the proposed method within the broader literature.


Weakness:

1. While the technical content is sufficient, the paper is not easy to read. It would benefit from a more global introduction and broader background context, rather than focusing almost entirely on the method itself. This would make the paper more accessible and easier to follow.

2. A key component of the proposed method—the point tracking (PT) model—is not clearly introduced or described. In particular, the paper lacks details about the PT model’s input, output, and working mechanism. Since this module is crucial for enabling joint optimization from RGB inputs, a brief explanation—perhaps as a preliminary section—would be helpful. As a reader, it was unclear how joint optimization could be performed using only raw RGB pixels until it became evident that the method relies on extracted point correspondences.

3. The proposed method targets dynamic videos, but it should also apply to static scenes. I am curious how it performs on standard static video benchmarks such as 7-Scenes, Tanks and Temples, or ScanNet. While it is not necessarily expected to outperform methods specifically designed for static scenes, such a comparison would serve as a useful reference and help assess the method’s generalizability.

4. It would be helpful to include the number of frames per scene when reporting runtime comparisons in Table 9 and other related tables. This would provide better context for evaluating the computational efficiency across different scenarios.

---

> ### Author Rebuttal · Authors · 2025-07-29
>
> # Response to Reviewer 9gnd:
> Thank you for your acknowledgment of our work. We highly appreciate your time and feedback, and address your concerns below.
>
> ## About Weakness 1, ‘Broader background context for a more readable paper’
> Thank you for your valuable suggestions. We appreciate your recognition of the technical content in our paper. As the current submission cannot be modified, we plan to enhance the background information in the camera-ready version to facilitate a clearer understanding of our work. In particular, we will provide a more comprehensive introduction to the research field at the beginning of our paper. Furthermore, regarding the pre-trained PT model that serves as the foundation for our proposed patch-wise tracking filters, we will include a more detailed explanation in the main paper—or, if permitted by the page limit, in a dedicated preliminary section.
>
> ## About Weakness 2, ‘Introduction about the pre-trained PT model’
> Thank you for pointing it out. We plan to add a more detailed explanation of the pre-trained PT model in the main paper in the camera-ready version. The default pre-trained PT model on which our proposed method is built is CoTracker. CoTracker is a transformer-based model achieving dense 2D point tracking in long video sequences. Unlike other related works, it jointly tracks the 2D points so that the dependencies on each other can help improve the accuracy and robustness of the model. During inference, the vanilla CoTracker allows a set of 2D query points and an RGB video as inputs and directly outputs the 2D location and visibility of each 2D query point on each frame. Although it significantly improved the PT performance compared with the existing works, it still suffers from the tracking location drift across frames, especially when one point disappears from the camera frustum and reappears in the field of view, as shown in Figure 9 in the Appendix. Such observations motivate us to propose the patch-wise tracking filters to extract only the robust, reliable, and sparse tracking trajectories as our pseudo-supervision.
>
> ## About Weakness 3, ‘Performance on static scene dataset’
> Thank you for your suggestions. We agree with your words that, 'although it is not necessarily expected to outperform methods specifically designed for static scenes, such a comparison would serve as a useful reference and help assess the method’s generalizability'. We conducted additional experiments on the Tanks and Temples dataset (Table 1 below). Due to the character limit, for other baselines, please refer to Table 2 in CF‑3DGS. Compared with CF‑3DGS (SOTA static-scene method that uses GT intrinsics), although our approach does not surpass the SOTA methods specialized for static scenes with extra supervision, such low errors can also support our generalizability.
>
>
> ### Table 1: Camera Pose Evaluation on static-scene Tanks and Temples dataset (ATE$\downarrow$/RPT trans$\downarrow$/RPE rot$\downarrow$)
> | Scenes | Ballroom | Barn | Church | Family |
> |--------|----------|------|--------|--------|
> | Ours   |  0.005/0.003/0.526  | 0.0188/0.006/0.915 |  0.037/0.003/0.301  |  0.021/0.002/0.440  |
>
> | Scenes  | Francis | Horse | Ignatius | Museum |
> |---------|---------|-------|----------|--------|
> | Ours | 0.063/0.010/0.883  |  0.012/0.007/0.823  |  0.028/0.008/0.628  |  0.008/0.020/0.551  |
>
> We summarize below the reasons why meaningful comparisons with existing works on the 7‑Scenes and ScanNet datasets are not feasible:
> 1) Access to the ScanNet dataset requires an application process and approval, which takes additional time.
> 2) After recheck, we found that the existing 7‑Scenes evaluations (e.g., DUSt3R / Fast3R) are based on unordered images and pairwise pose metrics. In contrast, our method fundamentally relies on long, continuous video sequences and temporal consistency. Since the official protocol merges multiple short sequences into unordered inputs and does not provide a single long sequence, it is not possible to conduct a fair or meaningful comparison on this dataset.
>
>
> ## About Weakness 4, ‘include the number of frames per scene when reporting runtime comparisons in Table 9 and other related tables’
>
> Thank you for your helpful suggestions. Since the experimental section of our paper has already demonstrated the high efficiency property of our method, and due to the space limits of rebuttal, we only present here the runtime of our method on different datasets along with the corresponding number of frames, as shown in the following Tables 2, 3, and 4. We plan to incorporate these additional results into the runtime tables in the camera-ready version to make it more complete.
>
> ### Table 2: Quantitative Runtime Results on DAVIS with Corresponding Frame Count
> | Metric  | Camel | Bear | Breakdance-flare | Car-roundabout | Car-shadow | Car-turn | Cows | Dog | Dog-agility |
> |---------|-------|------|------------------|----------------|------------|----------|------|-----|-------------|
> | Cam (min) | 1.57  | 3.15 | 1.73             | 4.97           | 0.93       | 2.97     | 2.85 | 1.60| 0.67        |
> | Frames  | 90    | 82   | 71               | 75             | 40         | 80       | 104  | 60  | 25          |
>
> | Metric  | Goat | Hike | Horsejump-high | Lucia | Motorbike | Parkour | Rollerblade | Tennis | Train |
> |---------|------|------|----------------|-------|-----------|---------|-------------|--------|-------|
> | Cam (min) | 2.10 | 4.20 | 1.97           | 1.97  | 2.08      | 9.07    | 1.25        | 3.47   | 1.90  |
> | Frames  | 90   | 80   | 50             | 70    | 43        | 100     | 35          | 70     | 80    |
>
> ### Table 3: Quantitative Runtime Results on iPhone with Corresponding Frame Count
> | Metric  | Apple | Paper-windmill | Space-out | Backpack | Block | Creeper | Handwavy |
> |---------|-------|----------------|-----------|----------|-------|---------|----------|
> | Cam (min) | 33    | 15             | 23        | 7        | 27    | 27      | 15       |
> | Frames  | 475   | 277            | 429       | 180      | 447   | 360     | 303      |
>
> | Metric  | Haru-sit | Mochi-high-five | Pillow | Spin | Sriracha-tree | Teddy | Wheel |
> |---------|----------|-----------------|--------|------|---------------|-------|-------|
> | Cam (min) | 10       | 6               | 21     | 30   | 15            | 31    | 27    |
> | Frames  | 200      | 180             | 330    | 426  | 302           | 472   | 437   |
>
> ### Table 4: Quantitative Runtime Results on NeRF-DS with Corresponding Frame Count
> | Metric  | Bell | As | Basin | Plate | Press | Cup | Sieve |
> |---------|------|----|-------|-------|-------|-----|-------|
> | Cam (h) | 1.05 | 0.95 | 0.75 | 0.53 | 0.68  | 1.02 | 0.78  |
> | Frames  | 881  | 846 | 668   | 424   | 487   | 807  | 584   |
>
>
> ## About Question 1, ‘Do other point trackers work as well?’
> Thank you for your question. As stated in Lines 141–143 of the main paper, although our proposed patch-wise tracking filters do an iterative process until there are $B$ tracked points in each frame, such filters do $\underline{not}$ conduct any $\underline{repeated}$ refinement or optimization of point tracking on the $\underline{same}$ tracking trajectory. This is the main reason why our runtime grows almost linearly with frame count, as shown in Figure 4 of the main paper. Thus, the pre-trained PT model on which our proposed filters are based must be a **feed-forward prediction model, outputting point locations and visibility**; we will clarify this further in the camera-ready version.
>
> In Table 8 of the main paper, we show ablation experiments on 2 different popular pre-trained PT models, CoTracker and CoTracker3. The results show that our proposed method is nearly independent of building on the particular PT models, as long as these requirements are met.
>
> Besides, Table 5 below summarizes all baseline PT methods reported in the CoTracker and CoTracker3 papers. CoTracker and CoTracker3 are the only two methods satisfying our requirements. Thus, other PT models cannot be incorporated in our proposed method due to architectural incompatibility.
>
> ### Table 5: Properties of All Baseline Methods Reported in CoTracker and CoTracker3.
> | Method                      | Type                          | Output Format                      |
> |-----------------------------|-------------------------------|------------------------------------|
> | TAP-Net            | Feed-forward                  | Point locations only               |
> | OmniMotion            | Optimization-based            | Point locations only               |
> | PIPs               | Iterative (sliding window)    | Point locations only               |
> | MFT               | Feed-forward                  | Point locations only               |
> | PIPs++           | Iterative refinement          | Point locations only               |
> | TAPIR            | Recurrent                     | Point locations + Visibility   |
> | **CoTracker**         | **Feed-forward**              | **Point locations + Visibility**   |
> | TAPTR                | Recurrent/transformer         | Point locations + Visibility       |
> | LocoTrack           | Recurrent                     | Point locations + Visibility       |
> | **CoTracker3**        | **Feed-forward**              | **Point locations + Visibility**   |
> | BootsTAPIR         | Recurrent                     | Point locations + Visibility       |
>
> We are also happy to discuss any other proper pre-trained PT models to explore potential improvements on our current proposed method. Please feel free to let us know.

---

> ### Author Response · Authors · 2025-08-03
> **Official Comment by Authors**
>
> We hope our rebuttal has addressed your questions and resolved your concerns regarding our work. Please let us know if there is anything else we can clarify. We would greatly appreciate a brief reply to share your thoughts. Thank you very much for your time.

---

> > ### Comment · Reviewer_9gnd · 2025-08-04
> >
> > My concerns have been almost addressed. Additionally, the 7-scenes dataset provides sequences with each containing 500-1000 frames. Is it too short for the proposed method? Hope authors can revise this and attach static scene results in the camera-ready version.

---

> > > ### Author Response · Authors · 2025-08-04
> > > **Official Comment by Authors**
> > >
> > > Thank you for your response! We are so glad to see our answers have addressed your concerns.
> > >
> > > ## About attaching the static scene results in the camera-ready version.
> > > Thank you for your suggestion. We will attach the static scene results in the camera-ready version.
> > >
> > > ## About whether 500-1000 frames are too short for the proposed method.
> > > Regarding the concern about video length, we believe that sequences with 500–1000 frames sufficiently demonstrate the effectiveness of our proposed method. As our answers to **Reviewer nBqP, About Question 3, ‘Testing on extremely long videos (>2000 frames)’**, we explored the maximum number of frames our method could handle on our device (a single NVIDIA A100 GPU (40 GB) with 256 GB of system RAM). Specifically, we selected two dynamic-scene sequences exceeding 2000 frames (footlab_3rd and scene_j716_3rd) from the Point Odyssey dataset for this evaluation. However, even after trimming the sequence to 1500 frames, our model still encounters RAM capacity limitations on our setup. We plan to enhance our model’s capacity to handle a greater number of frames on our current hardware setup.
> > >
> > > Thank you again!

---

> > > ### Author Response · Authors · 2025-08-05
> > >
> > > Thank you once again for your valuable time and commitment. We would like to kindly note that the discussion period will conclude in approximately 48 hours. If any points require further clarification, please don’t hesitate to reach out. If there is no remaining problem, we are looking forward to your final rating. Thank you!

---

> > > ### Author Response · Authors · 2025-08-06
> > >
> > > Thank you once again for your valuable feedback. Although the deadline was extended, we would like to kindly note that the discussion period will conclude in approximately 48 hours. We would like to try our best to address all your concerns. **If there is no remaining problem, we are looking forward to your final rating.**  **Thank you!**

---

> > > > ### Comment · Reviewer_9gnd · 2025-08-06
> > > >
> > > > My concerns have been addressed in the rebuttal, and I will keep my original rating unchanged.

---

> > > > > ### Author Response · Authors · 2025-08-07
> > > > >
> > > > > Thank you for your positive rating! We would like to kindly remind you to please submit the Mandatory Acknowledgment at your earliest convenience. Thank you again!

---

> ### Author Response · Authors · 2025-08-09
>
> Thank you for your acknowledgment of our paper. The discussion period will end in several hours. We would like to kindly remind you to submit the Mandatory Acknowledgment. Thank you!

---

### Official Review · Reviewer_izms · 2025-06-24

**Clarity:** 4
**Significance:** 4
**Originality:** 4
**Rating:** 6
**Confidence:** 3

**Summary:**

The paper proposes an RGB-only camera-parameter optimisation pipeline for casually captured dynamic-scene videos.
Key contributions are: 1) Patch-wise Tracking Filters, 2) Outlier-aware Joint Optimisation 3) A Two-Stage Optimisation Strategy.
Extensive experiments on 5 public datasets (NeRF-DS, DAVIS, iPhone, MPI-Sintel, TUM-dynamics) show better accuracy and up to 12× faster runtime than previous RGB-only baselines (casualSAM, COLMAP w/o mask) and, on some videos, even LiDAR-assisted Record3D. Ablations confirm each design choice.

**Questions:**

You state in line 273 that the performance is “nearly independent” of the PT backbone (Table 8).  Please add results with a much weaker tracker to support the claim, or consider improving the statement for rigorous discussion.

**Ethical Concerns:**

["NO or VERY MINOR ethics concerns only"]

**Final Justification:**

I have no further questions. The author’s explanation makes sense and does not affect my original rating.

**Limitations:**

yes

**Paper Formatting Concerns:**

no particular Paper Formatting Concerns

**Quality:**

4

**Strengths And Weaknesses:**

Strengths
（1）Well-designed optimisation objective; thorough ablations and evaluation on five datasets.
（2）Method is laid out step-by-step with helpful figures; loss derivations in appendix; datasets & metrics clearly listed.
（3）Addressing a practical bottleneck for 4D reconstruction; runtime improvement is compelling for real use.
（4）Novel coupling of sparse hinge-like tracks with Cauchy uncertainty in bundle adjustment; two-stage soft-plus trick is neat

Weaknesses

All designs are well motivated, and the overall presentation is of high quality. Minor weaknesses, like the Abrupt introduction of COLMAP and visual odometry at the beginning of the paper, which confuses my understanding of the target problems, are suggested to be improved.

---

> ### Author Rebuttal · Authors · 2025-07-29
>
> # Response to Reviewer izms:
> Thank you for your acknowledgment. We really appreciate your time and feedback, and address your concerns below.
>
> ## About Weakness 1, ‘Abrupt introduction on COLMAP and Visual Odometry’
> Thank you for your suggestions on paper writing. We plan to add more background information at the beginning of the paper, to help us make the introduction of visual odometry, and the state-of-the-art COLMAP more fluent.
>
>
> ## About Question 1, ‘Results with a much weaker tracker’
>
> Thank you for your question. As stated in Lines 141–143 of the main paper, although our proposed patch-wise tracking filters do an iterative process until there are $B$ tracked points in each frame, such filters do $\underline{not}$ conduct any $\underline{repeated}$ refinement or optimization of point tracking on the $\underline{same}$ point trajectory. This is also the main reason why our runtime grows almost linearly as frame count increases, as shown in Figure 4 of the main paper. Thus, the pre-trained PT model on which our proposed filters are based must be a **feed-forward prediction model, outputting point locations and visibility**; we will clarify this further in the camera-ready version.
>
> In Table 8 of the main paper, we show ablation experiments on 2 different popular pre-trained PT models, CoTracker and CoTracker3. The results show that our proposed method is nearly independent of building on the particular PT models, as long as these requirements are met.
>
> Besides, Table 1 below summarizes all baseline PT methods reported in the CoTracker and CoTracker3 papers. CoTracker and CoTracker3 are the only two methods satisfying our requirements. Thus, other PT models cannot be incorporated in our proposed method due to architectural incompatibility.
>
> ### Table 1: Properties of all baseline methods reported in the CoTracker and CoTracker3.
> | Method                      | Type                          | Output Format                      |
> |-----------------------------|-------------------------------|------------------------------------|
> | TAP-Net            | Feed-forward                  | Point locations only               |
> | OmniMotion            | Optimization-based            | Point locations only               |
> | PIPs               | Iterative (sliding window)    | Point locations only               |
> | MFT               | Feed-forward                  | Point locations only               |
> | PIPs++           | Iterative refinement          | Point locations only               |
> | TAPIR            | Recurrent                     | Point locations + Visibility   |
> | **CoTracker**         | **Feed-forward**              | **Point locations + Visibility**   |
> | TAPTR                | Recurrent/transformer         | Point locations + Visibility       |
> | LocoTrack           | Recurrent                     | Point locations + Visibility       |
> | **CoTracker3**        | **Feed-forward**              | **Point locations + Visibility**   |
> | BootsTAPIR         | Recurrent                     | Point locations + Visibility       |
>
> We are also happy to discuss any other proper pre-trained PT models to explore potential improvements on our current proposed method. Please feel free to let us know.

---

> ### Author Response · Authors · 2025-08-03
> **Official Comment by Authors**
>
> We hope our rebuttal has addressed your questions and resolved your concerns regarding our work. Please let us know if there is anything else we can clarify. We would greatly appreciate a brief reply to share your thoughts. Thank you very much for your time.

---

> ### Author Response · Authors · 2025-08-04
> **Official Comment by Authors**
>
> Thank you once again for your time and thoughtful feedback. As the discussion period concludes in three days, we remain fully available to address any remaining concerns or questions regarding our work. Please don’t hesitate to let us know if any further clarification is needed. We sincerely appreciate your engagement.

---

> ### Author Response · Authors · 2025-08-05
>
> Thank you once again for your valuable time and commitment. We would like to kindly note that the discussion period will conclude in approximately 48 hours. We greatly appreciate the opportunity to address any remaining questions or concerns you may have regarding our work. If any points require further clarification, please don’t hesitate to reach out. Thank you very much.

---

### Official Review · Reviewer_79BF · 2025-06-30

**Clarity:** 3
**Significance:** 2
**Originality:** 2
**Rating:** 4
**Confidence:** 3

**Summary:**

This paper proposes a SfM method, which estimates camera trajectory from an RGB video of the dynamic scenes. Unlike COLMAP and its successors, this approach doesn’t require any supervision and estimates camera intrinsics and extrinsic only from the RGB frames of the video. The method introduces three contributions: 1) patch-wise keypoint filter, which removes dynamic points and extracts stable points from the entire sequence, which will be used as a supervision signal; 2) Outlier handling mechanism integrated in the camera pose optimization procedure focusing on suppressing points coming from the dynamic objects; 3) Two step optimization strategy , which enhances the speed of the optimization.  The method is evaluated on several datasets demonstrating improved performance and greater efficiency compared to the   related approaches.

**Questions:**

$\Gamma_{raw}$ seems to be important parameter and its choice has not been experimentally analyzed in the main paper. It would be good to perform such analysis and backup the claims made in the two stage optimization procedure.

The qualitative results presenting camera trajectories in Fig.6 are good, but I would appreciate to see more examples in different scenes, especially, somehow ranked by the level of difficulties, e.g. from small to large numbers of moving outliers(low dynamic to high dynamic scenes).

It is strongly recommended to include and discuss failure cases both quantitatively and qualitatively (examples of recovered trajectories).

**Ethical Concerns:**

["NO or VERY MINOR ethics concerns only"]

**Final Justification:**

I checked the authors' replies to my questions and I appreciate their answers and effort they put in the explanations. I also checked other reviews and answers to them. Authors are strongly encouraged to include ablations on parameter choice and include a discussion about failure cases. I am happy about the rebuttal and will increase the rating to borderline accept.

**Paper Formatting Concerns:**

No major concerns

**Quality:**

3

**Strengths And Weaknesses:**

Strengths: The main strength of the proposed method is their computational efficiency achieved due to their two step optimization strategy. Moreover, paper achieves reasonable performance given that it doesn’t use any GT supervision. The experimental evaluation is very detailed and well done. The quality of the writing could be improved.

Weaknesses: The overall novelty looks fairly standard containing steps used in SfM methods. The choice of the steps is reasonable, but in spite the ablation study it is hard to assess how is each step critical in different dynamic scenes. For example, it would be good to comment how many correct point trajectories have been extracted after the first step in different dynamic scenes, starting from a low to high amounts of moving outliers. Further, the question is what amount of outliers can be handled by the proposed outlier handling strategy. The current proposed outlier handling strategy is similar to standard reweighed least square procedure combined with the robust estimators, e.g. Tukey estimator. It would be good to compare the performance with other robust estimators.
In the experimental evaluation it is evident that the methods with mask supervision or other GT supervision signals perform better then the proposed method. The question is how would the proposed method perform with such supervisions. It would be interesting to assess if the proposed optimization and outlier handling strategy would outperform the other methods if GT supervision was used.

---

> ### Author Rebuttal · Authors · 2025-07-29
>
> # Response to Reviewer 79BF:
> Thank you for your acknowledgment of our paper. We sincerely appreciate your time and feedback, and address your concerns below. As your concerns were not explicitly numbered or listed, we summarize them as follows.
>
> ## About Weakness 1, ‘how many correct point trajectories..xxx..starting from a low to high amounts of moving outliers.’
> Thank you for your question. We believe there must be a misunderstanding, and we should make it clear in the camera-ready version with the following clarifications.
>
> Firstly, we do not define point trajectories by correctness. Both inlier and outlier trajectories can be correct or incorrect, and these datasets lack GT optical flow or tracking labels, so such a definition would be meaningless. Instead, we classify trajectories into inliers and outliers, which directly motivates our proposed outlier-aware joint optimization.
>
> Secondly, the number of moving outliers does not affect trajectory accuracy. Our patch-wise tracking filters are built on CoTracker, which can robustly track trajectories regardless of the number of moving points, as discussed in the CoTracker paper.
> For further analysis on our model’s capacity to handle different amounts of outliers, please see our next response.
>
> ## About Weakness 2, ‘What amount of outliers can be handled?’
>
> Thank you for the question. By “amount of outliers,” we assume you mean the percentage of outliers among all trajectories. **We want to clarify that model performance depends not only on this percentage but also on the appearance pattern of outliers.** To illustrate, we provide a table comparing performance, outlier percentage, and appearance patterns across failure cases and one success case (motion masks for MPI-Sintel are manually annotated). As shown in Table 1 (below) and Fig. 21 (Appendix), a high outlier percentage (e.g., Ambush_6) can still yield good results, while lower percentages can fail when the appearance pattern is extreme. Thus, there is no strict threshold for the outlier percentage that leads to failure.
>
> ### Table 1: Failure Case Analysis by Outlier Percentage and Appearance Pattern on MPI-Sintel dataset.
> | Case Type            | Scene       | Outliers Percentage | ATE   | RPE trans | Appearance Pattern                                              |
> |----------------------|-------------|------------------------|-------|-----------|-----------------------------------------------------------------|
> | Failure  | cave_2      | 35.14%                 | 1.609 | 0.226     | Cover / Sweep across the frame    |
> |  Failure  | cave_4      | 39.36%                 | 0.125 | 0.035     | Dominate screen across the frame   |
> |  Failure  | market_5    | 34.31%                 | 2.027 | 0.617     | Dominate screen across the frame    |
> |  Failure | temple_3    | 31.92%                 | 0.782 | 0.1467    | Cover / Sweep across the frame |
> | Sampled Success | ambush_6    | 44.30%                 | 0.080 | 0.129     | Dominate half screen across the frame         |
>
> ## About Weakness 2, ‘Why not the Tukey estimator? Other estimator?’
> Thank you for asking this question. To our knowledge, we believe what you mean by ‘the Tukey estimator’ is Tukey’s biweight estimator. In the weighting function of Tukey’s biweight estimator,
> $$
> w(u) =
> \begin{cases}
> \left(1 - \left(\frac{u}{c}\right)^2\right)^2, & \text{if } |u| < c \\\\
> 0, & \text{if } |u| \geq c
> \end{cases}
> $$
> Tukey’s biweight completely discards residuals beyond the cutoff $c$, a hard threshold that must be tuned for each video. In contrast, our Cauchy weighting learns all parameters, allowing our outlier-aware joint optimization to generalize across videos.
>
> We are also happy to discuss other estimators to explore potential improvements on our current proposed method. Please feel free to let us know.
>
> ## About Weakness 3, ‘How would the proposed method perform with more GT supervisions?’
> Thank you for your question. More GT supervision (depth, camera pose, motion masks, 3D point clouds, camera intrinsic) will change the architecture of our pipeline, which is specifically designed for RGB-only supervision. So under such circumstances, our model needs to be carefully redesigned. But, theoretically, like most other models, we believe our method will perform better with more GT supervision. We might leave this as a future research path.
>
> Furthermore, we want to reclarify that our biggest contribution is our capability to **accurately and efficiently** estimate the camera parameters in dynamic scenes **solely supervised by a single RGB video, without any other GT supervision**.
>
> ## About Question 1, ‘Analysis on $\Gamma^{raw}$'
> Thank you for your suggestion. As shown in Table  7 of the main paper, replacing our two-stage strategy with a randomly initialized and learnable $\Gamma^{raw}$ leads to a clear performance drop.
> To address the request for a deeper analysis, we conducted additional ablations on $\Gamma^{raw}$ (see table below).
> '**Meaningless**' indicates that when $\Gamma^{raw}$ is frozen in Stage 1, other value settings behave the same as setting $\Gamma^{raw}=1$, so ablations 2 and 3 duplicate the settings of ablation 1 and the full model.
>
> ### Table 2: Analysis on $\Gamma^{raw}$ on NeRF-DS dataset. (Stage 1: 200 iterations; Stage 2: 50 iterations.)
> | Model              | Stage 1 Initialization                 | Stage 2 Initialization                 | PSNR  | SSIM   | LPIPS  |
> |--------------------|----------------------------------------|----------------------------------------|-------|--------|--------|
> | Full model         | $\Gamma^{raw}$ = 1 (freeze)                 | $\Gamma^{raw}$ = $\mathbb{E}^{ACP}_{stage1}$ (learn)                  | 33.55 | 0.9381 | 0.1182 |
> | More ablation 1    | $\Gamma^{raw}$ = 1 (freeze)                 | $\Gamma^{raw}$ = 1 (learn)                  | 31.81 | 0.9073 | 0.1403 |
> | Existing Ablation  | $\Gamma^{raw}$ = Random (learn)             | -  (learn)                    | 25.95 | 0.8100 | 0.2668 |
> | More ablation 2    | $\Gamma^{raw}$ = Random (freeze)            | -  (learn)             |          |    **Meaningless**    |        |
> | More ablation 3    | $\Gamma^{raw}$ = Random (freeze)            | $\Gamma^{raw}$ = $\mathbb{E}^{ACP}_{stage1}$ (learn)                  |       |   **Meaningless** |     |        |
>
> Such comparisons show that our initialization strategy on $\Gamma^{raw}$ in different stages is very important; otherwise, the performance drops dramatically.
>
> ## About Question 2, ‘more examples in different scenes, ranked by the level of difficulties, e.g., ..xxx... (low dynamic to high dynamic scenes)’
>
> Thank you for your questions. To clarify, the difficulty of camera evaluation depends on multiple factors (outliers, camera motion, appearance pattern, scene dynamics, etc.). Analysis on parts of them is in our answers to **About Weakness 2, ‘What amount of outliers can be handled?’**. To our knowledge, there are no explicit quantitative or qualitative metrics to rank it. We will clarify this in the camera-ready version.
>
> Unlike the related methods (Table 1 in the main paper), mostly supported by experiments on 30 - 50 videos, our results are supported by experiments on 5 popular public camera evaluation datasets covering about 70 videos across different difficulty levels. To make this clearer, we summarize the dataset characteristics (Section C of the Appendix) in the following table and will include it in the camera-ready version.
>
> ### Table 3: Dataset Features
> | Dataset       | Scene Dynamics Level | Scene Dynamics Pattern                                    | Camera Motion Level  | Camera Motion Pattern                                                                 |
> |---------------|----------------------|-----------------------------------------------------------|-----------------------|---------------------------------------------------------------------------------------|
> | NeRF-DS       | Large                | Shaking movement of reflective oject             | Large                 | Circle-like movement w/ large rotation change                 |
> | DAVIS         | Large                | Regular movement of daily common objects  | Large                 | Curve-like movement w/ proper rotation change              |
> | MPI-Sintel    | Large & Complex      | Cartoon characters’ irregular movement                   | Large                 | Straight-line-like movement w/ large rotation change          |
> | iPhone        | Large & Complex      | Irregular movement of daily common objects               | Large & Complex       | Fluctuating movement w/ large rotation change         |
> | TUM-Dyn  | Large & Complex      | Irregular movement of humans                              | Large & Complex       | Fluctuating movement with large rotation change      |
>
> Due to the character limit, please refer to our response to **Reviewer  9gnd, About Weakness 3, ‘Performance on static scene dataset,’** for a further investigation of the generalization ability of our method. Thank you.
>
> ## About Question 3, ‘More qualitative and quantitative analysis on failure cases’
> Thank you for your advice. We will add the following discussions to the paper in the camera-ready version. As shown in Fig. 21 (Appendix, L586–589), our method fails when videos are dominated by large, fast-moving objects (e.g., humans, dragons). We attribute it to two main reasons: (1) a high proportion of outliers and (2) extreme outlier patterns where large objects sweep and cover the screen. These factors hinder robust, sparse hinge-like relations extracted for accurate pseudo-supervision. More details about the reasons can be seen in our answers to **About Weakness 2, ‘What amount of outliers can be handled?’**.

---

> > ### Comment · Reviewer_79BF · 2025-08-08
> >
> > I am satisfied with your answers and do not have further questions.

---

> > > ### Author Response · Authors · 2025-08-08
> > >
> > > Thank you for your response! We sincerely appreciate your positive acknowledgment of our paper after considering our rebuttal. It is gratifying to know that our responses have successfully addressed your concerns.

---

> ### Author Response · Authors · 2025-08-03
> **Official Comment by Authors**
>
> We hope our rebuttal has addressed your questions and resolved your concerns regarding our work. Please let us know if there is anything else we can clarify. We would greatly appreciate a brief reply to share your thoughts. Thank you very much for your time.

---

> ### Author Response · Authors · 2025-08-04
> **Official Comment by Authors**
>
> Thank you once again for your time and thoughtful feedback. As the discussion period concludes in three days, we remain fully available to address any remaining concerns or questions regarding our work. Please don’t hesitate to let us know if any further clarification is needed. We sincerely appreciate your engagement.

---

> ### Author Response · Authors · 2025-08-05
>
> Thank you once again for your valuable time and commitment. We would like to kindly note that the discussion period will conclude in approximately 48 hours. We greatly appreciate the opportunity to address any remaining questions or concerns you may have regarding our work. If any points require further clarification, please don’t hesitate to reach out. Thank you very much.

---

> ### Author Response · Authors · 2025-08-07
>
> Thank you for your valuable feedback. Although the deadline has been somewhat extended, we would like to kindly remind you that the discussion period will end in approximately 48 hours. If you have any more questions, please let us know ASAP so that we have enough time to address your concerns. If you believe our rebuttal can help address your concerns, we sincerely expect you to consider improving your score as the final rating. Thank you!
>
>
> PS: Please do not forget to submit the Mandatory Acknowledgement.

---

> ### Author Response · Authors · 2025-08-08
>
> Thank you for your reviews! We would like to kindly remind you that the discussion period will end in approximately 24 hours. If there are any more concerns, please let us know so that we can address them ASAP. We also sincerely invite you to see our discussions with other reviewers if you share similar concerns with them. We sincerely expect you to consider improving your score as the final rating. Thank you again!

---

### Official Review · Reviewer_nBqP · 2025-07-02

**Clarity:** 3
**Significance:** 3
**Originality:** 3
**Rating:** 4
**Confidence:** 3

**Summary:**

This paper solves an important problem in dynamic scenes “solely supervised by RGB video”. To solve this problem, three key components are proposed as the main contributions of this paper.

**Questions:**

1. How did you determine w=12/24 and τ_var=0.1? Can you provide sensitivity analysis for these parameters? What happens when there are insufficient high-texture patches?

2. You mention the method fails with large moving objects. Can you quantify this - what percentage of moving pixels causes failure? Is there a way to predict these failure cases?

3. Have you tested sequences longer than 900 frames ? What’s about the stability ? For example, 2000 frames, since COLMAP failed occasionally.

**Ethical Concerns:**

["NO or VERY MINOR ethics concerns only"]

**Final Justification:**

Overall, I hold a positive evaluation of this paper, and the rebuttal phase has successfully addressed all of my concerns. I agree with reviewer 79BF's point that "The overall novelty looks fairly standard containing steps used in SfM methods," but this paper indeed tackles a rather challenging problem: "RGB-only supervised camera parameter optimization in dynamic scenes." The authors have conducted thorough comparisons with previous work and demonstrated the superiority of their method.

What I am not entirely satisfied with in this paper is that the numbers in the figures are too blurry (due to being saved in non-vector format), which causes some reading difficulties and diminishes the overall aesthetic quality of the paper. Therefore, I maintain my weak accept recommendation and suggest that the authors revise the figures.

**Limitations:**

yes

**Quality:**

3

**Strengths And Weaknesses:**

**Strengths:**

1. Well-structured methodology: The paper presents a comprehensive approach with three well-motivated components that address specific challenges in RGB-only camera parameter optimization for dynamic scenes.

2. Extensive experiments: The authors conduct thorough experiments across 5 datasets with both quantitative metrics and qualitative visualizations,  I appreciate authors' hard work.

3. Efficiency improvements: The method achieves substantial runtime reductions (e.g., 55% of the second-fastest method on NeRF-DS) while maintaining or improving accuracy, which is practically valuable.

4. Ablation studies: The paper provides detailed ablation studies validating each component's contribution to the overall performance.

**Weakness:**

1. It seems all the words on the figures are blured, this reduces readability and aesthetics. May the authors make the words on figures clearly ? For example, change all of the figures to vector format.

2. The claim of being the "first RGB-only supervised" method (Contribution 1, line 61) contradicts Table 1, which clearly shows existing RGB-only methods like Robust-CVD and casualSAM. This should be corrected to accurately position the work's contributions.

3. The patch-wise tracking filters appear to be incremental improvements over existing point tracking methods (CoTracker). I think there should be deeper analysis like:

- How patch size $w$ affects performance across different scene types
- Sensitivity analysis of the texture threshold $τ_{var}$

4. I encourage authors present limitation in the maintext. I think there should be more discussion when and why the method fails.

---

> ### Author Rebuttal · Authors · 2025-07-29
>
> # Response to Reviewer nBqP:
> Thank you for your acknowledgment of our paper. We sincerely appreciate your time and feedback, and address your concerns below.
>
> ## About Weakness 1, ‘The blurry words in the figures’
> Thanks for your suggestions. After re-checking, we found that some figures were not correctly saved as vector graphics. Since the article cannot be modified at this stage, we plan to correct it in the camera-ready version.
>
> ## About Weakness 2, ‘Words’ contradictory’
> Thank you for pointing out this misunderstanding. What we meant is that our method is the first RGB-only supervised approach for dynamic scenes that is both accurate and efficient, as existing RGB-only methods are either slow or inaccurate. We will remove the word “first” in the camera-ready version.
>
> ## About Weakness 3, ‘How patch size affects performance across different scene types?’
> Thank you for raising this question. **We would like to clarify that although patch size can influence performance, the scene type is not a factor in this process; thus, we do not adjust patch size based on scene types.** Instead, as noted in Line 200 of our main paper, the patch size $w$ is determined solely by the frame size.
>
> Specifically, when the input aspect ratio (long/short side) exceeds 1.5, we consider the image elongated. Camera motion in such cases causes obvious uneven scene changes across axes, so we adopt a smaller patch size to capture more valid high-texture patches. Otherwise, we use a larger patch size. The value of $w$ is chosen as a trade-off between memory cost and efficiency, **but it should definitely be adjusted for extremely small or large frame sizes**. We will add such discussions to the camera-ready version. Further sensitivity analysis on $w$ is provided in our response to **About Question 1, ‘Sensitivity analysis on $w$ and $\tau_{var}$’**.
>
> ## About Weakness 3, ‘Sensitivity analysis of the texture threshold’
> Thank you for your advice. Please see our subsequent answers to **About Question 1, ‘Sensitivity analysis on $w$ and $\tau$’**, for a detailed discussion about the sensitivity analysis of the texture threshold.
>
> ## About Weakness 4, ‘The position of limitations’
> Thank you for your suggestions. We will move the limitation section to the main paper in the camera-ready version.
>
> ## About Weakness 4, ‘More discussion on when and why the method fails?’
> Thank you for your advice. We should discuss more in the camera-ready version. As shown in Fig. 21 (Appendix, L586–589), our method fails when videos are dominated by large, moving humans and dragons. We attribute it to two main reasons: 1) high outlier percentage; 2) extreme outlier patterns where large objects sweep and cover the screen. These factors hinder robust, sparse hinge-like relations extracted as accurate pseudo-supervision. More details can be seen in our answers to **About Question 2, ‘What percentage of moving pixels causes failure? ..xxx.. failure cases?’**.
>
> ## About Question 1, ‘How did you determine $w$=12/24 and $τ_{var}$=0.1?’
> Thank you for your question. Please refer to our answers to **About Weakness 3, ‘How patch size affects performance across different scene types?’** for a thorough analysis of $w$.
>
> **We sincerely apologize for the typo in the paper. The value “$τ_{var}$=0.1” should be corrected to “$τ_{var}$=0.01”.** We will correct this typo in the camera-ready version.
>
> We empirically set $\tau_{var}=0.01$ and use this fixed value for all videos. Eq. 1 in main paper selects relatively high-texture patches by keeping those whose intensity variance exceeds $0.01$ $\times$ $\sigma*$ ($\sigma*$ is the maximum patch variance in a frame). This removes textureless regions (e.g., plain backgrounds, white walls; see Fig. 9 in Appendix) while retaining useful information, as only patches with variance below 1% of the maximum are discarded.
>
> ## About Question 1, ‘Sensitivity analysis on $w$ and $\tau_{var}$’
>
> Thank you for asking this question. We conduct the sensitivity analysis on these two parameters, as shown in the subsequent two tables with detailed discussions. We plan to add the following two tables to the ablation study section of our main paper in the camera-ready version to make our paper more complete.
>
>
> ### Table 1: Sensitivity Analysis on Patch Size $w$ on NeRF-DS dataset
> | Scene Optimization | Camera Optimization           | PSNR  | SSIM   | LPIPS  |
> |--------------------|-------------------------------|-------|--------|--------|
> | 4DGS               | ours($w$=12, default)     | 33.55 | 0.9381 | 0.1182 |
> |  4DGS               |   ours($w$=16)               | 32.23 | 0.9085 | 0.1411 |
> |  4DGS               |    ours($w$=24)              | 24.71 | 0.7722 | 0.3084 |
>
> ---------------------------------------------------------
>
> ### Table 2: Sensitivity Analysis on Variance Threshold $\tau_{var}$ on NeRF-DS dataset
> | Scene Optimization | Camera Optimization                 | PSNR                                   | SSIM                                   | LPIPS                                  |
> |--------------------|-------------------------------------|----------------------------------------|----------------------------------------|----------------------------------------|
> | 4DGS          | Ours($\tau_{var}$ = 0.01, default)        | 33.55                                  | 0.9381                                 | 0.1182                                |
> |   4DGS        | Ours($\tau_{var}$ = 0.04)                        | 32.45                   | 0.9136                | 0.1367               |
> |    4DGS       | Ours($\tau_{var}$ = 0.08)                          | 32.14                   | 0.9016                | 0.1408                           |
> |   4DGS       | Ours($\tau_{var}$ = 0.16)                          | Fail                                                       |
>
> Tables 1 and 2 show that our model is more sensitive to $w$ than $\tau_{var}$. Such ablation studies also demonstrate our good value definition on these two parameters. However, too large $\tau_{var}$ makes the model fail due to a lack of useful information.
>
> ## About Question 1, ‘What happens when there are insufficient high-texture patches?’
> Thank you for asking this question. To clarify, what we mean by ‘high-texture patches’ is the **relatively** high-texture patches. We should make it clearer in the camera-ready version. Like our answers to **About Question 1, ‘How did you determine w=12/24 and τ_var=0.1?’** above, in each frame, we only discard the relatively low-texture patches whose intensity variance is less than 1% of the maximum intensity variance. This efficiently avoids losing useful high-texture patches. However, we might share limitations with others in certain extreme scenarios, such as videos that predominantly feature plain walls with super-limited texture. We will leave it as our future research topic.
>
> ## About Question 2, ‘What percentage of moving pixels causes failure? Is there a way to predict these failure cases?’
> Thank you for asking this question. **We would like to clarify that there is no strict threshold for the percentage of moving pixels that leads to failure.** Because the model’s ability to handle outliers is also affected by the appearance pattern of outliers. Table 3 below compares the performance, outlier percentage, and appearance pattern in all failure cases and one sampled success case. We plan to add this table to the main paper of the camera-ready version. Since the MPI-Sintel dataset does not provide any GT motion mask, we manually annotate the motion masks.
>
> ### Table 3: Failure Case Analysis by Outlier Percentage and Appearance Pattern
> | Case Type            | Scene       | Percentage of outliers | ATE   | RPE trans | Appearance Pattern                                              |
> |----------------------|-------------|------------------------|-------|-----------|-----------------------------------------------------------------|
> | Failure case        | cave_2      | 35.14%                 | 1.609 | 0.226     | Object briefly sweeps across the frame, temporarily covering it |
> |  Failure case  | cave_4      | 39.36%                 | 0.125 | 0.035     | Object persistently dominates most of the frame.               |
> |  Failure case  | market_5    | 34.31%                 | 2.027 | 0.617     | Object persistently dominates most of the frame.               |
> |  Failure case  | temple_3    | 31.92%                 | 0.782 | 0.1467    | Object briefly sweeps across the frame, temporarily covering it |
> | Sampled Success | ambush_6    | 44.30%                 | 0.080 | 0.129     | Object persistently dominates about half of the frame.         |
>
> As shown in the above table and Figure 21 of the Appendix, the model’s performance is not strictly determined by the proportion of outliers: in ambush_6, the model performs well even with a large number of outliers, while in others, performance degrades despite a small proportion of outliers. The appearance pattern also affects the performance of the model.
>
> Currently, there is no quantitative indicator available to predict whether the method will succeed or fail. But we highly suggest forecasting based on the appearance pattern.
>
> ## About Question 3, ‘Testing on extremely long videos (>2000 frames)’
> Thank you for your questions. To our knowledge, existing test datasets and prior methods do not provide extremely long videos (>2000 frames) nor report experiments under such conditions. So we select two sequences exceeding 2000 frames (footlab_3rd and scene_j716_3rd) from the Point Odyssey dataset. However, even after trimming the sequence to 1500 frames, our model still encounters RAM capacity limitations on our setup, where all experiments are conducted on a single NVIDIA A100 GPU (40 GB) with 256 GB of system RAM. The longest video in our current experiments contains 881 frames (bell in NeRF-DS). We plan to investigate methods for reducing GPU and RAM consumption as part of our future work.

---

> ### Author Response · Authors · 2025-08-03
> **Official Comment by Authors**
>
> We hope our rebuttal has addressed your questions and resolved your concerns regarding our work. Please let us know if there is anything else we can clarify. We would greatly appreciate a brief reply to share your thoughts. Thank you very much for your time.

---

> ### Author Response · Authors · 2025-08-04
> **Official Comment by Authors**
>
> Thank you once again for your time and thoughtful feedback. As the discussion period concludes in three days, we remain fully available to address any remaining concerns or questions regarding our work. Please don’t hesitate to let us know if any further clarification is needed. We sincerely appreciate your engagement.

---

> ### Author Response · Authors · 2025-08-05
>
> Thank you once again for your valuable time and commitment. We would like to kindly note that the discussion period will conclude in approximately 48 hours. We greatly appreciate the opportunity to address any remaining questions or concerns you may have regarding our work. If any points require further clarification, please don’t hesitate to reach out. Thank you very much.

---

> ### Author Response · Authors · 2025-08-07
>
> Thank you for your valuable feedback. Although the deadline has been somewhat extended, we would like to kindly remind you that the discussion period will end in approximately 48 hours. If you have any more questions, please let us know ASAP so that we have enough time to address your concerns. If you believe our rebuttal can help address your concerns, we sincerely expect your final rating. Thank you!
>
>
> PS: Please do not forget to submit the Mandatory Acknowledgement.

---

> > ### Comment · Reviewer_nBqP · 2025-08-08
> > **Comment of rebuttal**
> >
> > Thank you for the authors' detailed response. I have carefully read the authors' replies as well as their responses to other reviewers' questions, and I have no further questions. This paper's contributions have certain practical significance, therefore I will maintain my rating and strongly recommend the authors to improve the presentation of qualitative results for readability and aesthetics.

---

> > > ### Author Response · Authors · 2025-08-08
> > >
> > > Thank you for your reply and for acknowledging our paper. It is our pleasure to address all of your concerns, and we look forward to receiving your final positive rating. Additionally, we would like to kindly remind you to submit the Mandatory Acknowledgment. Thank you again!

---

> ### Author Response · Authors · 2025-08-09
>
> Thank you for your acknowledgment of our paper. The discussion period will end in several hours. We would like to kindly remind you to submit both the final rating and the Mandatory Acknowledgment. Thank you!

---

### Note · Authors · 2025-08-12

Dear NeurIPS 2025 Reviewers, ACs, SACs, and PCs,

We sincerely appreciate the reviewers’ recognition of our novel work and their valuable efforts during the review process. It has been our pleasure to engage in constructive discussions, address all concerns, and maintain (Reviewer nBqP, izms, 9gnd) / receive (Reviewer 79BF) positive acknowledgements in the reviewers' response after rebuttal.

In summary, while existing works often rely on additional priors to improve the accuracy and efficiency of camera estimates in dynamic scenes, **we investigate whether it is possible to achieve comparable or even superior results using only RGB supervision—the most minimal form of supervision**. Motivated by such a question, we proposed our novel RGB-only supervised, accurate, and efficient camera parameter optimization method in dynamic scenes, including 3 key components: (1) Patch-wise Tracking Filters, to establish robust and maximally sparse hinge-like relations across the RGB sequence. (2) Outlier-aware Joint Optimization, for efficient camera parameter optimization by adaptive down-weighting of moving outliers, without reliance on motion priors. (3) A Two-stage Optimization Strategy, to enhance stability and optimization speed by a trade-off between the Softplus limits and convex minima in losses.

We also conducted extensive qualitative and quantitative experiments on 5 public datasets, including approximately 70 videos in total, as presented in the paper. Additionally, we present further experiments on a static-scene dataset during rebuttal, as requested by the reviewers. All experiments show that our method is currently the most accurate, efficient, and RGB-only supervised dynamic-scene camera parameter estimation method.

To show the effectiveness and contribution of our proposed method, within the paper, we conducted adequate ablation experiments about each module and our new design. During rebuttal, we provide more thorough ablation studies following the request of reviewers on parameter values, the percentage of outliers, and $\Gamma^{raw}$.

We also analyze our failure cases and others together quantitatively and qualitatively in the main paper and in the rebuttal.

Thank you all for your time and effort during the whole review and rebuttal period. Thank you all for your positive acknowledgement of our work. Every feedback, response, and comment helps us make our paper better and better. Thank you again!

Best Regards,

The authors of Paper 13570

---

### Decision · Program_Chairs · 2025-09-17

**Decision:**

Accept (spotlight)

**Comment:**

This paper introduces an RGB-only supervised method for camera parameter estimation in dynamic scenes, combining patch-wise tracking filters, outlier-aware optimization, and a two-stage strategy. Reviewers praised the clear methodology, thorough ablations, and strong runtime and accuracy across ~70 videos, with some noting novelty is moderate and presentation could improve. The rebuttal addressed concerns by clarifying claims, adding sensitivity and static-scene experiments, and committing to improved figures and explanations. Overall, the work is solid, practical, and broadly relevant. I recommend acceptance.